# Learning from Highly Sparse Spatio-temporal Data

**Leyan Deng[†], Defu Lian [†§*], Chenwang Wu[†], Enhong Chen[†§]**

[†] School of Artificial Intelligence and Data Science, University of Science and Technology of China
[§] School of Computer Science and Technology, University of Science and Technology of China
{dleyan, wcw1996}@mail.ustc.edu.cn, {liandefu, cheneh}@ustc.edu.cn

## Abstract

Incomplete spatio-temporal data in the real world has spawned much research. However, existing methods often utilize iterative message-passing across temporal and spatial dimensions, resulting in substantial information loss and high computational cost. We provide a theoretical analysis revealing that such iterative models are susceptible to data and graph sparsity, causing unstable performances on different datasets. To overcome these limitations, we introduce a novel method named One-step Propagation and Confidence-based Refinement (OPCR). In the first stage, OPCR leverages inherent spatial and temporal relationships by employing a sparse attention mechanism. These modules propagate limited observations directly to the global context through one-step imputation, which is theoretically affected only by data sparsity. Following this, we assign confidence levels to the initial imputations by correlating missing data with valid data. This confidence-based propagation refines the separate spatial and temporal imputation results through spatio-temporal dependencies. We evaluate the proposed model across various downstream tasks involving highly sparse spatio-temporal data. Empirical results indicate that our model outperforms state-of-the-art imputation methods, demonstrating its effectiveness and robustness.

## 1 Introduction

Spatio-temporal data, encompassing information distributed across both spatial and temporal dimensions, is prevalent in various domains [1]. In real world, the acquisition of spatio-temporal data often faces practical constraints, leading to missing observations. Incomplete data undermines the reliability and effectiveness of subsequent tasks, necessitating robust imputation techniques. However, the existing spatio-temporal works effort addresses temporal missing (i.e., point-level missing), which is usually caused by inevitable device failure. Beyond that, spatial missing (i.e., node-level missing) becomes a burning issue. For example, in a traffic scenario, the management will not deploy dense traffic sensors to reduce costs. Therefore, exploring spatially sparse data not only makes the best use of available data but also lowers the barriers to implementing spatio-temporal models in real scenarios. In view of the gap, Traffic4cast 2022 (T4C22) competition [2] presented sparse traffic data collected in real urban road networks. Specifically, the competition considered three cities, where sensors are deployed sparsely, i.e. spatial and point missing co-exist. T4C22 aims at generalizing limited data to entire city to predict global traffic dynamics.

We first provide a theoretical analysis for general spatio-temporal iterative imputation model from PAC-learnability perspective. Then we find that iterative methods are constrained by the sparsity and structure of data. As the data scale and missing rate increase, the model requires more iterations. Additionally, with an increasing number of iterations, the model requires more training samples. Therefore, for highly sparse large-scale data, iterative methods suffer from information loss and error accumulation. To address these issues, we propose a sparse attention-based one-step imputation and confidence-based refinement approach. In the first stage, we propagate information from observed

data to all missing data directly, leveraging inherent spatial and temporal correlations. In the second stage, we assign confidence-based propagation weights to the imputed results. Through confidence-based refinement, we eliminate the bias introduced by imputations from separate perspectives.

This paper makes the following contributions.

- We provide an limitation analysis for the popular iterative imputation from the PAC-learnability perspective, which explains the error accumulation caused by multiple iterations.

- Motivated by theoretical results, we propose a one-step propagation strategy to efficiently recover global information from limited observations. Then we assign confidence-based propagation weights to spatio-temporal interactions, refining the imputation results.

- We apply our method to various downstream tasks, the experiments show that our model can learn sufficient information from limited observations than state-of-the-art baselines.

## 2   Related Works

There are many works exploring how to impute different types of sparse data. For tabular data, matrix factorization-based methods are widely popular. A recent work transformed tabular data imputation into link prediction in a bipartite graph, which benefited from the expressive power of graph neural networks (GNNs). For graph data, a simple idea is propagating known features to missing data through the graph structure. For example Feature propagation (FP) [3] iteratively aggregates neighboring node features and PCFI [4] further considered pseudo-confidence to propagate messages more accurately. It is clear that GNN can capture spatial relations better. In addition to GNN-based imputation model, SAT [5] and ITR [6] both additionally considered structural representations and structure reconstruction objective. However, these graph imputation problems focus on static graph and fixed missing data sets, which is incompatible with dynamic sparse spatio-temporal data.

For time-series data, the existing approaches investigated the imputation performances of different sequence models. For example, BRITS [7] proposed a bidirectional Recurrent Neural Networks (RNN). SAITS [8] designed self-attention based model and jointly trained imputation and reconstruction objective. CSDI [9] proposed a conditional score-based diffusion model. However, these time-series imputation methods ignore intrinsic spatial dependencies in spatio-temporal data.

For spatio-temporal data, existing research leverages topological information effectively. IGNNK [10] introduced a scalable inductive method trained by reconstructing random sub-graphs. GRIN [11] was the first to apply Graph Neural Networks (GNNs) to multivariate time-series imputation and proposed a graph-based recurrent neural network. Subsequently, SPIN [12] identified that auto-regressive models like GRIN are prone to error propagation. SPIN developed an end-to-end architecture utilizing intra-node and inter-node attention mechanisms to address this. However, when dealing with highly sparse large-scale data, propagating valid information through graph structures requires multiple iterations, leading to error accumulation and information loss. In addition, PriSTI [13] proposed a diffusion-based model. Nonetheless, PriSTI employs two separate attention modules to incrementally aggregate temporal and spatial dependencies, which leads to decoupling the spatio-temporal context. Additionally, PriSTI uses linear interpolation for coarse conditional information, which is inadequate for spatially missing data. Beyond attribute imputation, PoGeVon [14] also addresses the challenge of missing structural information in spatio-temporal data.

## 3   Preliminaries

**Definition 3.1 (Spatio-temporal Series).** Given a fixed graph $G = (\mathcal{V}, \mathcal{E}, \boldsymbol{A})$, where $\mathcal{V}$ and $\mathcal{E}$ denote the set of $N$ nodes and $E$ edges, respectively; $\boldsymbol{A} \in \mathbb{R}^{N \times N}$ is the corresponding adjacency matrix. A spatio-temporal series of length $T$ is represented by $\boldsymbol{X} \in \mathbb{R}^{N \times T \times F_x}$, where $F_x$ denotes the number of feature dimensions. We represent each spatio-temporal point (ST point) with a tuple $z = (v, t)$ and $\boldsymbol{x}_{v,t} \in \mathbb{R}^{F_x}$ denotes the collected data of node $v$ at time $t$.

**Definition 3.2 (Spatio-temporal Mask).** For spatio-temporal series $\boldsymbol{X}$, we use a binary mask $\boldsymbol{M}$ to indicate missing ST points, where $m_{v,t} = 0$ if $\boldsymbol{x}_{v,t}$ is missing, otherwise $m_{v,t} = 1$. Note that, since a faulty device often miss all records, we ignore the availability of data in feature dimension. Thus,

we can define a unique sparse spatio-temporal series $\tilde{X}$ in terms of $X$ and $M$, i.e.,

$$\tilde{X} = X \odot M + NaN \odot (1 - M), \tag{1}$$

**Definition 3.3** (**Sparse Spatio-temporal Data Learning**). The goal is to accomplish downstream tasks based on the incomplete data $\tilde{X}$. Without loss of generality, we consider the following objective functions for node-level task.

$$\mathcal{L}(Y, \hat{Y}) = \frac{1}{N} \sum_{v \in \mathcal{V}} l(y_v, \hat{y}_v), \tag{2}$$

where $l(\cdot, \cdot)$ is an element-wise loss function that depends on specific downstream tasks. Here $y_v$ and $\hat{y}_v$ are the ground truth and model's prediction for node $v$, respectively.

For imputation task, the model aims to minimize the reconstruction error as follows.

$$\mathcal{L}(\tilde{X}, \hat{X}) = \frac{1}{|1 - M|} \sum_{v \in \mathcal{V}} \sum_{t \in \mathcal{T}} (1 - m_{v,t}) \cdot l(x_{v,t}, \hat{x}_{v,t}), \tag{3}$$

where $\mathcal{T}$ denotes the set of time steps and $l(\cdot, \cdot)$ is an element-wise loss function.

## 4 Theoretical Analysis

As we stated above, the iterative spatio-temporal imputation methods will suffer from information loss and error accumulation on extremely sparse large-scale data. In order to expose the limitations of iterative models, we provide a theoretical analysis in terms of PAC-learnability for node-level tasks.

We use $X \in \mathcal{X}$ and $Y \in \mathcal{Y}$ to denote complete spatio-temporal series and node-level labels, and use $M \in \mathcal{M}$ to denote mask distribution. Let $\mathcal{D}$ be the fixed but unknown distribution over $\mathcal{X} \times \mathcal{Y}$. We define the mixture distribution $\tilde{\mathcal{D}}$ of $\mathcal{D}$ and $\mathcal{M}$ using Eq. 1. Given a training set $\tilde{S} = \{(\tilde{X}_i, Y_i)\}^m$, we assume that all samples in $\tilde{S}$ are i.i.d. according to $\tilde{\mathcal{D}}$, denoted as $\tilde{S} \sim \tilde{\mathcal{D}}^m$. Let $f \in \mathcal{F} : \mathcal{X} \times \mathcal{M} \to \mathcal{Y}$ be a sparse spatio-temporal data learning model, the empirical risk over $\tilde{S}$ and the corresponding generalization risk is defined as follows,

$$\textbf{Empirical risk} : \hat{\mathcal{R}}_{\tilde{S}}(f) = \frac{1}{m} \sum_{i=1}^{m} l(f(\tilde{X}_i), Y_i),$$

$$\textbf{Generalization risk} : \mathcal{R}_{\tilde{D}}(f) = \mathbb{E}_{(\tilde{X}, Y) \sim \tilde{D}} \left[ l(f(\tilde{X}), Y) \right]$$

Similarly, we denote empirical and generalization risk under complete data as $\hat{\mathcal{R}}_S(f)$ and $\mathcal{R}_{\mathcal{D}}(f)$.

**Definition 4.1** (**PAC-Learnability**[15]). A concept class $\mathcal{C}$ is said to be probably approximately correct (PAC) learnable if there exist a learning algorithm $\mathcal{A}$ and a polynomial function $poly(\cdot, \cdot, \cdot, \cdot)$ satisfying: for any $\epsilon, \delta > 0$ and any distribution $\mathcal{D}$, the following holds for any sample size $m \geq poly(1/\epsilon, 1/\delta, n, size(c))$:

$$\mathbb{P}_{S \sim \mathcal{D}^m} [R_{\mathcal{D}}(h_S) \leq \epsilon] \geq 1 - \delta.$$

Most of sparse spatio-temporal data learning models have the following encoder-decoder architecture. The encoder $f_e$ learns all ST point representations from sparse data and the decoder $f_d$ outputs node-level predictions for specific downstream task.

$$f_d(f_e(\tilde{X})) = \phi \left( \frac{1}{T} \sum_{t \in \mathcal{T}} f_e(\tilde{X})_t W_d \right),$$

where $W_d$ represents the trainable weight matrix in decoder. In the encoder, the iterative models to learn the missing ST point representations work by continuously extracting valid information from its neighbors. The key differences between iterative models lie in how they define neighbors and how they aggregate messages from these neighbors.

Let $\mathcal{Z} = \{(v, t) | v \in \mathcal{V}, t \in \mathcal{T}\}$. For any ST point $z \in \mathcal{Z}$, we denote the set of its neighboring ST points at the $k$-th iteration by $\mathcal{N}_z^k$. According to the definition of neighbors, we can infer the path

between any two ST points. Without loss of generality, for any $z \in \mathcal{Z}$, we can formulate the $k$-th iteration as follows.

$$\boldsymbol{h}_z^k = \phi \left( \sum_{z' \in \mathcal{N}_z^k} p_{z' \to z}^k \cdot \boldsymbol{h}_{z'}^{k-1} \right),$$

where $\boldsymbol{h}_z^k$ denotes the learned representation for ST point $z$ after the $k$-th iteration. For message-passing, the iterative models usually assign weight $p_{z' \to z}^k$ to messages from $z'$ to $z$. We denote the distance of the path from ST point $z$ to $z'$ as $d_{z \to z'}$. Let $\mathcal{Z}_o \subseteq \mathcal{Z}$ to be the set of all observed ST points and $\mathcal{Z}_m = \mathcal{Z} \backslash \mathcal{Z}_o$. To ensure that the model has ability to recover all ST points, the number of iterations must satisfies $K \geq \max_{z' \in \mathcal{Z}_m} \min_{z \in \mathcal{Z}_o} d_{z \to z'}$. Therefore, multiple iterations are inevitable in highly sparse large-scale spatio-temporal data.

We then establish the PAC-learnable guarantee for the iterative spatio-temporal imputation methods. First, let us immediately mask some mild assumptions that are easy to implement.

**Assumption 4.2.** For the weight matrix in decoder, we assume its spectral norm satisfies $\|\boldsymbol{W}_d\|_2 \leq B_d$, for any ST point $z \in \mathcal{Z}$, we assume its feature vector satisfies $\|\boldsymbol{x}_z\|_2 \leq B_x$. For the loss function, we assume $l$ is an $C_l$-lipschitz continuous and bounded by $[0, 1]$. For the activation function, we assume $\phi$ is $C_\phi$-lipschitz continuous with $\phi(0) = 0$. For the mask distribution $\mathcal{M}$, we assume that all ST points are randomly masked with a probability of $\rho$.

**Proposition 4.3.** *Let $\mathcal{F}$ be a $K$-iterations imputation model class, we assume its learned propagation weights are bounded by $[0, \gamma]$. If we draw a sample $\tilde{S}$ of size $m$, for any $f \in \mathcal{F}$, the following inequality holds.*

$$\mathbb{P}_{\tilde{S} \sim \tilde{\mathcal{D}}^m} \left[ R_{\tilde{\mathcal{D}}}(h_{\tilde{S}}) \leq \epsilon \right] \geq 1 - \delta$$

*with $m \geq \frac{\log 1/\delta}{2\epsilon - 4\eta \mathcal{C} K - 4(1-\eta)\mathcal{C} K \cdot \rho^\tau - 4C_l \Re_m(\mathcal{F})}$, where $\mathcal{C} = C_l \cdot \gamma^K \cdot C_\phi^{K+1} \cdot (d_{\max})^K \cdot B_x B_d$.*

*Remark* 4.4. Here $0 < \eta < 1$, $d_{\max} = \max_{z \in \mathcal{Z}} |\mathcal{N}_z|$ and $\tau = \max_{z \in \mathcal{Z}} |\{d_{z' \to z} < K | z' \in \mathcal{Z}\}|$. Consider PAC-learnability under complete data, number of required samples needs to satisfy $m \geq \frac{\log 1/\delta}{2(\epsilon - 2C_l \Re_m(\mathcal{F}))}$. Since $\Re_m(\mathcal{F})$ represents Rademacher complexity of model class $\mathcal{F}$, it is encouraging that the learnability of the model on sparse data is related to the one on complete data. Thus, we can estimate the impact of the missing data on the model performance. Obviously, the required sample size $m$ is positively correlated with the missing ratio $\rho$ and the number of model iterations $K$. Nevertheless, the structural sparsity in the spatio-temporal data itself limits the learnability of the iterative model in addition to the data sparsity. Specifically, $\tau$ is related to the number of $k$-hop neighbors of each ST point. Therefore, the sparser the spatio-temporal structure (i.e., smaller $\tau$), the greater the number of samples required. This theoretical result provides an interpretation for iterative model-induced error accumulation. Note that although we assume that all ST points are masked at random (MAR), this proof strategy can be generalized to other mask distribution $\mathcal{M}$.

# 5    Methodology

This section presents our proposed method, One-step Propagation and Confidence-based Refinement (OPCR), as illustrated in Fig. 1. In the first stage, we independently learn intrinsic representations for ST points in both spatial and temporal dimensions. We then employ a sparse attention-based one-step propagation strategy to obtain two separate imputation results. In the second stage, we derive imputation confidence based on the learned correlations between ST points to address potential biases in these results. Finally, we perform confidence-based spatio-temporal propagation to refine the final predictions.

## 5.1    Sparse Attention-based One-step Propagation

Motivated by the aforementioned theoretical findings, we propose a one-step propagation method to avoid error accumulation and information loss commonly found in iterative methods. Specifically, as illustrated in Fig. 1, we begin by independently learning intrinsic spatial and temporal representations for ST points. We capture correlations among ST points across different dimensions by adopting a sparse attention mechanism. This approach enables the limited information to be propagated directly to the global context as efficiently as possible.

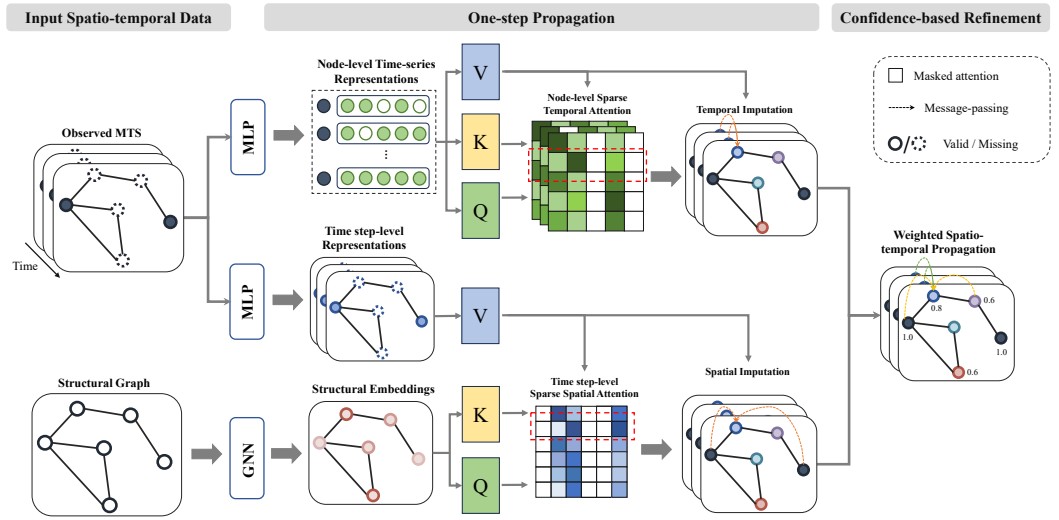

Figure 1: The framework of the proposed OPCR.

### 5.1.1 Sparse Spatial Attention

We first address the significant spatial sparsity present in spatio-temporal data. To fully utilize limited observation, we learn the correlations between nodes using static spatial information. Leveraging these intrinsic dependencies, messages from all observations are propagated directly to the global context through a sparse attention mechanism. We utilize a $L_s$-layers Graph Neural Network (GNN) [16] to learn the spatial representations of nodes from the graph structure. We initialize $\boldsymbol{S}_0$ using the inherent static attributes of all nodes; then the layer-wise updation can be summarized as follows.

$$\boldsymbol{S}_l = \phi \left( \boldsymbol{S}_{l-1} \boldsymbol{W}_1^l + \boldsymbol{D}^{-1} \boldsymbol{A} \boldsymbol{S}_{l-1} \boldsymbol{W}_2^l \right),$$

where $\boldsymbol{W}_1^l$ and $\boldsymbol{W}_2^l$ are the parameter matrix in the $l$-layer; $\boldsymbol{D}$ is the degree matrix of $\boldsymbol{A}$.

In order learn the spatial associations between nodes, we employ a self-attention mechanism [17]. We use node embeddings $\boldsymbol{S} = \boldsymbol{S}_{L_s}$ as both the key and query. We denote the set of available nodes by $\tilde{\mathcal{V}} \subset \mathcal{V}$. For any node $u \in \mathcal{V}, v \in \tilde{\mathcal{V}}$, the spatial self-attention is formulated as:

$$\boldsymbol{Q}^s, \boldsymbol{K}^s, \boldsymbol{V}_t^s = \boldsymbol{S} \boldsymbol{W}_Q^s, \boldsymbol{S} \boldsymbol{W}_K^s, \boldsymbol{X}_t \boldsymbol{W}_V^s, \tag{4}$$

$$\alpha_{u,v}^s = \frac{\exp(\langle \boldsymbol{q}_u^s, \boldsymbol{k}_v^s \rangle)}{\sum_{v' \in \tilde{\mathcal{V}}} \exp(\langle \boldsymbol{q}_u^s, \boldsymbol{k}_{v'}^s \rangle)},$$

where $\boldsymbol{Q}^s, \boldsymbol{K}^s, \boldsymbol{V}_t^s \in \mathbb{R}^{N \times F}$ represent the query, key, value of spatial self-attention; $\boldsymbol{q}_i^s, \boldsymbol{k}_i^s$ and $\boldsymbol{v}_{i,t}^s$ stand for their $i$-the row; $\boldsymbol{W}_Q^s, \boldsymbol{W}_K^s, \boldsymbol{W}_V^s \in \mathbb{R}^{F \times F}$ are parameter matrices. For any ST point $(v, t)$, we aggregate spatial messages coming from all observations weighted by learned attentions. The spatial dependencies-based imputation result can be computed as:

$$\boldsymbol{h}_{v,t}^s = \text{MLP} \left( \sum_{v' \in \tilde{\mathcal{V}}} \alpha_{v,v'}^s \boldsymbol{v}_{v',t}^s \right). \tag{5}$$

### 5.1.2 Sparse Temporal Attention

Another component is designed to learn temporal correlations. We adopt a temporal attention mechanism for the time-series data associated with each node. Unlike recurrent sequence models such as RNN or LSTM, Transformers are not inherently capable of learning sequence information[18]. We first introduce vanilla positional encoding [17] to capture this information. If real-world timestamps are available, we use a learnable embedding layer to encode these timestamps [19].

Combining positional encoding with timestamp encoding, for any node $v \in \mathcal{V}$, the input to the temporal sparse attention module can be formulated as follows.

$$\overline{\boldsymbol{X}}_v = \boldsymbol{X}_v + PE(\boldsymbol{X}_v) + MLP(\boldsymbol{U}),$$

$$PE_{pos,2i} = sin\left(pos/10000^{2i/d_{model}}\right)$$

$$PE_{pos,2i+1} = cos\left(pos/10000^{2i/d_{model}}\right)$$

where $\boldsymbol{X}_v$ is associated time-series of node $v$; $\boldsymbol{U}$ is the available real-world time information, such as the hour of the day.

For any node $v$, $\overline{\boldsymbol{X}}_v$ serves as the key, query, and value in the temporal self-attention mechanism. For node $v$, similar to the spatial module, we denote the available time steps by $\tilde{\mathcal{T}}_v$. Then the temporal attention score and the temporal aggregation are formulated as:

$$\boldsymbol{Q}_v^t, \boldsymbol{K}_v^t, \boldsymbol{V}_v^t = \overline{\boldsymbol{X}}_v \boldsymbol{W}_Q^t, \overline{\boldsymbol{X}}_v \boldsymbol{W}_K^t, \overline{\boldsymbol{X}}_v \boldsymbol{W}_V^t \quad (6)$$

$$(\alpha_{t_1,t_2}^t)_v = \frac{\exp(\langle \boldsymbol{q}_{v,t_1}^t, \boldsymbol{k}_{v,t_2}^t \rangle)}{\sum_{t_2' \in \tilde{\mathcal{T}}_v} \exp(\langle \boldsymbol{q}_{v,t_1}^t, \boldsymbol{k}_{v,t_2'}^t \rangle)},$$

$$\boldsymbol{h}_{v,t}^t = \mathrm{MLP}\left(\sum_{t' \in \tilde{\mathcal{T}}_v} (\alpha_{t,t'}^t)_v \boldsymbol{v}_{v,t'}^t\right),$$

where $\boldsymbol{Q}_v^t, \boldsymbol{K}_v^t, \boldsymbol{V}_v^t \in \mathbb{R}^{T \times F}$ represent the query, key, value of spatial self-attention; $\boldsymbol{q}_{v,i}^t, \boldsymbol{k}_{v,i}^t$ and $\boldsymbol{v}_{v,i}^t$ stand for their $i$-the row; $\boldsymbol{W}_Q^t, \boldsymbol{W}_K^t, \boldsymbol{W}_V^t \in \mathbb{R}^{F \times F}$ are parameter matrices.

Based on these two sparse attention modules, we propagate all available information to the global missing data in one step. We also give a PAC-Learnability analysis for the proposed method.

**Proposition 5.1.** *Let $\mathcal{F}$ be a sparse attention-based imputation model class. If we draw a sample $\tilde{S}$ of size $m$, for any $f \in \mathcal{F}$, the following inequality holds.*

$$\mathbb{P}_{\tilde{S} \sim \tilde{\mathcal{D}}^m}\left[R_{\tilde{\mathcal{D}}}(h_{\tilde{S}}) \leq \epsilon\right] \geq 1 - \delta$$

*with $m \geq \frac{\log 1/\delta}{2[\epsilon - 4\rho\mathcal{C} - 2C_l \Re_m(\mathcal{F})]}$, where $\mathcal{C} = C_l \cdot C_\phi \cdot B_d \cdot B_v$.*

*Remark* 5.2. It can be seen that the number of required samples is positively correlated with the sparsity $\rho$. Unlike iteration-based models, sparsity is the sole factor that constrains the performance of the sparse attention-based model when dealing with incomplete data. Consequently, the proposed one-step strategy is scalable to various spatio-temporal data of different sizes and structures.

## 5.2 Confidence-based Iterative Refinement

In the first stage, we reconstructed the missing representations based on inherent spatial and temporal information. However, these two independent modules decompose the global context. Integrating these results and refining them through spatio-temporal structure is a straightforward idea. To further propagate reliable information, we propose a confidence-based message-passing mechanism. Intuitively, there are correlations between any pairwise spatio-temporal points. Thus, the missing data will remove some dependencies. The larger the correlations between an ST point and other observations, the more plausible its recovery. Therefore, for any ST point $(u, t)$, we can define the confidences of its spatial and temporal imputation results as:

$$\beta_{u,t} = \frac{\sum_{v \in \tilde{\mathcal{V}}} \exp(\langle \boldsymbol{q}_u^s, \boldsymbol{k}_v^s \rangle)}{\sum_{v \in \mathcal{V}} \exp(\langle \boldsymbol{q}_u^s, \boldsymbol{k}_v^s \rangle)} + \frac{\sum_{t' \in \tilde{\mathcal{T}}_v} \exp(\langle \boldsymbol{q}_t^t, \boldsymbol{k}_{t'}^t \rangle)}{\sum_{t' \in \mathcal{T}} \exp(\langle \boldsymbol{q}_t^t, \boldsymbol{k}_{t'}^t \rangle)} \quad (7)$$

We extract highly confident parts from separate imputation results in temporal and spatial dimensions. However, simple weighting ignores the global context in spatio-temporal data. Therefore, we propose to employ the learned confidences as propagation weights through the spatio-temporal dimension. For any ST point $(u, t)$, we first define the set of its neighbors as

$$\mathcal{N}_{u,t} = \{(v, t) | v \in \mathcal{N}_u\} \cup \{(v, t') | t' \in \mathcal{T} \backslash t\},$$

where $\mathcal{N}_u$ denotes the set of neighboring nodes of node $u$ in the fixed graph $G$. Then the layer-wise message-passing for ST point $(u, t)$ can be formulated as

$$\boldsymbol{O}_{u,t}^{l+1} = \text{MLP}\left(\boldsymbol{O}_{u,t}^{l}\|\sum\nolimits_{(v,t')\in\mathcal{N}_{u,t}}\beta_{v,t'}\cdot\boldsymbol{O}_{v,t'}^{l}\right),$$

where $\boldsymbol{O}_{u,t}^0 = \boldsymbol{h}_{u,t}^s + \boldsymbol{h}_{u,t}^t$. During $L$ layers, we finally recover representations for all ST points. Then the decoder is designed depend on different downstream tasks.

## 6 Experiments

We evaluate our approach on three sets of real-world datasets for different downstream tasks, including imputation and prediction. More experimental details and time complexity analysis are provided in the appendix. The source code and datasets are available at `https://github.com/dleyan/OPCR`.

### 6.1 Datasets

We consider three sets of spatio-temporal datasets and summarize their statistics in Table 1. It is evident that these datasets vary in terms of topology size. The T4C22 dataset primarily comprises spatially missing data, exhibiting an exceptionally high sparsity of up to 90%.

Table 1: Statistics of the datasets.

|  | TRAFFIC | | LARGE-SCALE | | T4C22 | | |
|---|---|---|---|---|---|---|---|
|  | PEMS-BAY | METR-LA | PV-US | CER-E | LONDON | MADRID | MELBOURNE |
| # NODES | 325 | 207 | 5166 | 6435 | 59110 | 63397 | 49510 |
| # EDGES | 2369 | 1515 | 71446 | 51428 | 132414 | 121902 | 94871 |
| # STEPS | 52128 | 34272 | 8688 | 8688 | 10560 | 10464 | 10176 |

To simulate incomplete data in realistic scenario, we design two policies to inject missing data into original datasets: 1) `Point Missing`, in which we follow the same setup of [12, 11], randomly dropping $\rho$ of the available data. 2) `Spatial missing`, in which we are inspired by the T4C22 dataset, randomly dropping 25% of the available data, then mask out $\rho$ of the devices.

### 6.2 Baselines

We compare the proposed OPCR with various baselines designed for different data types.

- **Tabular Imputation.** We consider three statistical methods for matrix imputation: 1) `Mean`, imputing missing data using the mean value at each time step. 2) `Matrix Factorization` (`MF`) [20] with rank = 10. 3) `GRAPE` [21], a graph-based feature imputation method, which regards observations and features as two types of nodes in a bipartite graph, and the observed feature values as edges.

- **Graph Imputation.** For spatial missing, we consider some node attributes imputation methods: 1) `Feature Propagation` (`FP`) [3], iteratively propagating observed messages through graph structure. 2) `PCFI` [4], further measuring the propagation confidence.

- **Time-series Imputation.** To impute multivariate time-series, we consider: 1) `BRITS` [7], a bidirectional RNN-based model. 2) `SAITS` [8], a transformer-based model. 2) `CSDI` [9], a diffusion-based model.

- **Spatio-temporal Data Imputation**. We also consider some state-of-the-art methods for spatio-temporal data imputation. 1) `IGNNK` [10], an inductive GNN-based model. 3) `SPIN-H` [12], an efficient version of spatio-temporal attention based method. 3) `PriSTI` [13], a conditional diffusion model. 4) `PoGeVon` [14] , solving a specific issue in which spatio-temporal data contains missing values in both node time series features and graph structures.

### 6.3 Spatio-temporal Data Imputation Task

For the imputation task, we consider mean absolute error (MAE) as evaluation metrics. All the experiments are run with 5 different random seeds. In each round, we inject $\rho$ of missing data into

Table 2: Imputation performances (in terms of MAE) on Traffic dataset and Large-scale dataset .

| Model | Point Missing | | | | Spatial Missing | | | |
|---|---|---|---|---|---|---|---|---|
| | PEMS-BAY | METR-LA | PV-US | CER-E | PEMS-BAY | METR-LA | PV-US | CER-E |
| Mean | $5.00 \pm 0.00$ | $10.28 \pm 0.02$ | $8.34 \pm 0.00$ | $0.56 \pm 0.00$ | $5.08 \pm 0.13$ | $10.37 \pm 0.21$ | $8.45 \pm 0.28$ | $0.56 \pm 0.01$ |
| MF | $5.25 \pm 0.00$ | $7.55 \pm 0.04$ | $4.53 \pm 0.01$ | $0.50 \pm 0.00$ | $5.71 \pm 0.07$ | $7.61 \pm 0.11$ | $12.47 \pm 0.24$ | $0.57 \pm 0.01$ |
| GRAPE | $4.12 \pm 0.02$ | $6.75 \pm 0.03$ | $10.69 \pm 0.22$ | $0.34 \pm 0.00$ | $4.72 \pm 0.02$ | $6.78 \pm 0.03$ | $11.67 \pm 0.03$ | $0.36 \pm 0.00$ |
| FP | $4.92 \pm 0.00$ | $9.22 \pm 0.02$ | $7.60 \pm 0.01$ | $0.52 \pm 0.00$ | $5.05 \pm 0.05$ | $9.25 \pm 0.16$ | $7.87 \pm 0.13$ | $0.52 \pm 0.02$ |
| PCFI | $5.14 \pm 0.01$ | $12.94 \pm 0.03$ | $8.79 \pm 0.02$ | $0.52 \pm 0.01$ | $6.75 \pm 0.59$ | $14.95 \pm 1.86$ | $12.31 \pm 0.30$ | $0.50 \pm 0.04$ |
| BRITS | $2.58 \pm 0.00$ | $5.25 \pm 0.10$ | $30.48 \pm 4.75$ | $1.38 \pm 0.99$ | $5.64 \pm 0.16$ | $8.88 \pm 0.52$ | $25.77 \pm 9.00$ | $1.15 \pm 0.28$ |
| SAITS | $2.44 \pm 0.01$ | $5.21 \pm 0.03$ | $6.40 \pm 4.56$ | $0.67 \pm 0.57$ | $6.14 \pm 0.24$ | $9.51 \pm 0.63$ | $2.73 \pm 0.08$ | $0.37 \pm 0.11$ |
| CSDI | $2.16 \pm 0.04$ | $3.48 \pm 0.09$ | $7.51 \pm 2.30$ | $0.49 \pm 0.02$ | $3.76 \pm 0.75$ | $4.51 \pm 0.45$ | $8.97 \pm 1.93$ | $0.47 \pm 0.02$ |
| IGNNK | $2.61 \pm 0.01$ | $4.37 \pm 0.02$ | $7.86 \pm 0.02$ | $0.38 \pm 0.00$ | $4.70 \pm 0.13$ | $6.85 \pm 0.27$ | $11.43 \pm 0.06$ | $0.46 \pm 0.00$ |
| SPIN-H | $1.84 \pm 0.02$ | $2.99 \pm 0.03$ | $1.94 \pm 0.06$ | $0.33 \pm 0.09$ | $4.93 \pm 0.04$ | $7.62 \pm 0.06$ | $2.63 \pm 0.06$ | $0.29 \pm 0.00$ |
| PoGeVon | $5.68 \pm 0.01$ | $8.86 \pm 0.01$ | / | / | $5.73 \pm 0.14$ | $8.82 \pm 0.56$ | / | / |
| PriSTI | $2.05 \pm 0.02$ | $3.85 \pm 0.18$ | $11.93 \pm 2.84$ | $0.63 \pm 0.12$ | $3.05 \pm 0.20$ | $5.04 \pm 0.75$ | $11.75 \pm 7.54$ | $0.59 \pm 0.14$ |
| OPCR | $\mathbf{1.79 \pm 0.01}$ | $\mathbf{2.80 \pm 0.02}$ | $\mathbf{1.73 \pm 0.06}$ | $\mathbf{0.28 \pm 0.00}$ | $\mathbf{2.27 \pm 0.07}$ | $\mathbf{3.20 \pm 0.10}$ | $\mathbf{2.01 \pm 0.11}$ | $\mathbf{0.28 \pm 0.00}$ |

the entire dataset using different masking policies. All models are trained on the sparse training set and evaluated on the corresponding sparse testing set.

### 6.3.1 Highly Sparse Data Imputation

The experimental results of all models are summarized in Table 2, where the missing rate $\rho$ is set as 95%. There is a clear gap in the imputation performance on the traffic dataset between spatial-missing data and point-missing data, indicating that recovering spatial-missing data is a challenging issue. For the tabular and graph imputation baselines, their performance remains relatively consistent across datasets with spatial and point missing. This can be easily understood, as these baselines only utilize or fail to utilize spatial structure. The time-series imputation baselines are not suitable for spatial missing, as they solely rely on temporal information. In addition, it is difficult for them to model such a high-dimensional time series on large-scale datasets.

For state-of-the-art (SOTA) spatio-temporal series imputation baseline SPIN-H, they are most suitable for small-scale data with point missing. Since they only propagate partial observation iteratively, larger topology and higher spatial missing rate imply a need for more iterations. Therefore, the error accumulation and information loss during iterative process lead to their poor performance. For the inductive baseline IGNNK, it converts the original task to smaller-scale spatio-temporal series via sampling sub-graphs. The effectiveness of IGNNK proves that there are correlations between all nodes, which is consistent with our motivation. However, this sampling strategy still ignores some of the valid information. Therefore, considering the problem of error accumulation in autoregressive methods, some diffusion-based models have been proposed, such as CSDI [9] and PriSTI [13]. It is still challenging for CSDI to model a high-dimensional time series in the large-scale dataset. However, it show performance improvement on the traffic dataset compared to other temporal baselines (i.e., BRITS and SAITS). For the spatio-temporal method PriSTI, it achieves better performances than SPIN-H on the traffic dataset with spatial missing. However, its performances are unstable on the large-scale datasets, especially the spatially missing data. PoGeVon [14] is proposed to deal with specific data where attribute-missing and structure-missing co-exist. Therefore, it show worse imputation performances than other spatio-temporal baselines. In addition, PoGeVon does not apply to the large-scale dataset because of its high computational complexity.

As a result, our proposed OPCR outperforms all the baselines in most cases. Especially, on traffic dataset with spatial missing, OPCR exceeds the best baseline by 26% and 37%. on large-scale PVUS dataset, OPCR exceeds the best baseline by 11% and 24% under two settings. These experiments demonstrate the effectiveness and scalability of our proposed OPCR.

### 6.3.2 Imputation with Increasing Missing Rate

To evaluate the effect of data sparsity on imputation performance, we simulate sparse data at various rate, ranging from 25% to 95%. Figures 2 and 3 report imputation performances on point missing data and spatially missing data, respectively. On the traffic dataset, it is clear that the imputation errors increase with the missing rate. Similar to the results in Table 2, spatial missing poses a greater

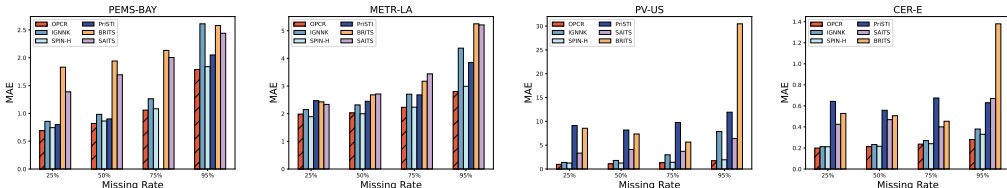

Figure 2: Imputation performance (in terms of MAE) with increasing point missing rate.

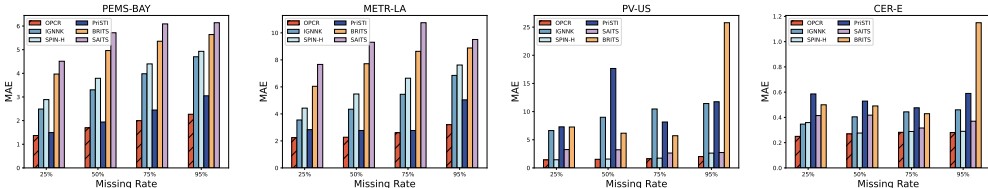

Figure 3: Imputation performance (in terms of MAE) with increasing saptial missing rate.

challenge for all models. On large-scale datasets, temporal baselines struggle to model extremely high-dimensional data and do not exhibit the same trends as observed in the traffic dataset. In addition, these models usually terminates training process early then result extremely poor imputation performances. For the inductive spatio-temporal baseline IGNNK, its overall performance show a significant improvement when the missing rate is reduced from 95% to 75%. This empirical results are due to IGNNK can not sample enough sub-graph data for training in highly-sparse data. For SPIN-H, its imputation error on the CER-E dataset is not exactly positively correlated with the missing rate, which is consistent with our theoretical findings. The diffusion-based method PriSTI achieves significant improvements over other baselines on the traffic dataset with spatial missing. However, PriSTI struggle to deal with large-scale dataset. For the proposed OPCR, we can observe that it outperforms baselines in most cases. Furthermore, the imputation performances of our method steadily improves with decreasing missing rates. This experimental result is consistent with the theoretical findings, confirming that the PAC learnability of the proposed OPCR is only constrained by data sparsity.

## 6.4 Traffic Prediction Task

For the T4C22 dataset, we follow the competition rules. We set the evaluation metric to a weighted cross-entropy loss for the congestion classification task, and MAE for the prediction task. All the models are equipped with the same downstream network and use the same training strategy. Table 3 presents the traffic performance comparison for two downstream tasks. Since we designed strong downstream models for both prediction tasks, the performance gap is insignificant. It can be seen that our proposed OPCR outperforms baselines across all experiments. There are also some findings in the traffic prediction experiments. SPIN-H performs poorly compared to graph imputation baselines (i.e., FP and PCFI). This is because the T4C22 dataset only considers short-term (i.e., 4 time steps) historical series, and its highly spatial sparsity results in slow propagation of valid information.

Table 3: Traffic Prediction performances.

| MODEL | CONGESTION CLASSIFICATION | | | TRAVEL TIME PREDICTION | | |
|---|---|---|---|---|---|---|
| | LONDON | MADRID | MELBOURNE | LONDON | MADRID | MELBOURNE |
| FP | 0.8126 | 0.8624 | 0.8635 | 84.56 | 68.17 | 31.48 |
| PCFI | 0.8159 | 0.8514 | 0.8552 | 105.61 | 61.06 | 31.43 |
| SPIN-H | 0.8335 | 0.8768 | 0.8795 | 77.21 | 59.24 | **31.36** |
| **OPCR** | **0.8114** | **0.8444** | **0.8519** | **74.44** | **55.39** | 31.54 |

## 6.5 Ablation Study

To evaluate the effectiveness of different components of our proposed OPCR, we make comparisons between some model variants. To demonstrate the robustness of our model, we also conduct sensitivity

analysis with respect to model's hyper-parameters. All models here are trained on the traffic datasets and large-scale datasets with 95% missing rate.

**Effectiveness of Each Component.** We consider the following variants. 1) `Spatial module (S)`: Only use spatial module as encoder. 2) `Temporal module (T)`: Only use temporal module as encoder. 3) `W/O refinement (W/O R)`: Ignore the confidence-based refinement.

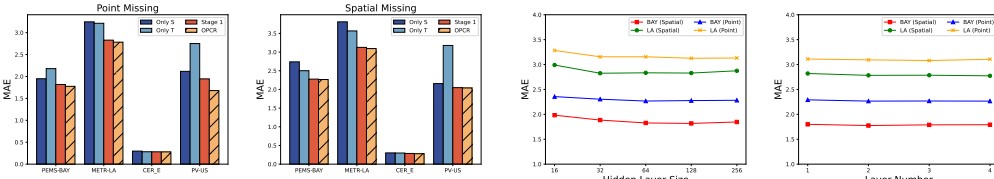

Figure 4: The effect of each component.          Figure 5: The effect of hyper-parameters.

The results of the ablation study are shown in Fig. 4. First, we compare the spatial module and temporal module. Obviously, it is challenging for both variants to handle spatially missing data. On the large-scale dataset, the temporal module presents poorer performances compared to the spatial module. This result is easy to understand because the temporal module utilizes a narrower range of valid information. Second, combining temporal-based and spatial-based results yields superior performances compared to a single module. Finally, when comparing the complete model with all variants, it can be seen that confidence-based propagation also plays a vital role.

**Sensitivity w.r.t. Hyper-parameters.** First, we consider the effect of hidden states, as shown in Fig. 5. As the hidden states go from 16 to 256, the overall performance slowly increases and then fluctuates slightly. Therefore, we set the number of hidden states to 128. This result reflects the robustness of the proposed model. Then, we consider the sensitivity w.r.t. the number of layers in stage 2, as shown in Fig. 5. The approximate best level can be reached when the number of layers is only 2. These experiments prove that our design motivation is that a shallow iterative propagation is enough to refine the complete recovery by the first stage.

# 7   Conclusion

For the highly sparse spatio-temporal data learning problem prevalent in the real-world, the existing common scheme is iterative propagation-base imputation. However, such methods empirically suffer from error accumulation and high computational costs in large-scale data. Therefore, we first provide a PAC-learnability analysis for iterative imputation methods from the perspective of PAC-learnability, which theoretically demonstrates their limitations. Motivated by theoretical findings, we propose one-step propagation and confidence-based refinement (OPCR). In the first stage, we rapidly propagate useful information to all missing data through a sparse attention mechanism that makes full use of limited observations. To eliminate the bias caused by independent temporal and spatial modeling, we propose assigning confidence to imputation results and achieving more accurate spatio-temporal message-passing. We evaluate the imputation and prediction performances of the proposed OPCR on a dataset of different scales. The experimental results illustrate our approach outperforms the state-of-the-art imputation methods. There are several interesting future directions. First, it would be interesting to explore some downstream tasks without supervision. The second is to extend our theoretical analysis to non-random missing scenarios.

# 8   Acknowledgment

The work was supported by grants from the National Natural Science Foundation of China (No. 92367110).

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

# A Experiment

## A.1 Datasets

We consider the following three sets of spatio-temporal datasets.

- **Traffic dataset**[22] record traffic dynamics every 5 minutes in San Francisco Bay Area and LA County, respectively [11].
- **Large-scale dataset**[23] includes two larger-scale datasets. The sources of the PV-US dataset and the CER-E dataset are the simulated energy production by PV farms [1] and the CER Smart Metering Project [2], respectively. Both datasets are aggregated in 30 minutes.
- **T4C22 dataset** [24] record traffic dynamics every 15 minutes from vehicle detectors in three cities, which is provided by Traffic4cast 2022 competition [3].

For the traffic dataset and PV-US dataset, we derive the adjacency from pairwise geographic distances [22, 25]. Without location information, the adjacency of the CER-E dataset is derived from similarity between historical series. For the T4C22 dataset, the available urban road networks can serve as adjacency.

## A.2 Experimental settings

Considering memory consumption, for SPIN, we choose the efficient version SPIN-H as baseline. On the traffic dataset, we train all models with an RTX 3090 GPU (24GB RAM). On the large-scale dataset and the T4C22 dataset, we train them with a V100 GPU (16GB RAM). All the baselines have been implemented in PyTorch [26]. We use, whenever possible, the open-source code and configuration provided by the authors.

For spatio-temporal data imputation task, we select windows of length $24$. On the traffic datasets, we strictly follow the settings of SPIN[4]. We fix the maximum number of epochs to 300 and we use early stopping on the validation set with patience of 40 epochs. On the large-scale dataset, considering memory capacity and computational efficiency, we reduced the number of hidden states to 16 for some baselines (i.e., SPIN-H, CSDI, and PriSTI). We fix the maximum number of epochs to 200 and we use early stopping on the validation set with patience of 10 epochs.

For traffic prediction task, we follow the competition settings. Since the original test dataset is unlabeled, we randomly select data from the training set for different days. Note that this partitioning strategy is aligns with the competition rules. We provided each model with the same downstream network and trained the models end-to-end. For congestion classification task, we train all the models for 20 epochs. For travel time prediction task, we train all the models for 50 epochs. We both select the best model with the minimum loss on the validation set for evaluation.

## A.3 Time Complexity and Run-time Analysis

We compare our proposed OPCR with SOTA iterative imputation baselines SPIN. We let the number of edges in $G$ be $E$ and the number of iterations for all models be $L$. According to [12], the time complexity of SPIN scales with $\mathcal{O}\left(L(N+E)T^2\right)$ and its efficient version SPIN-H scales with $\mathcal{O}\left(L(N+E)KT\right)$, where $K \ll T$.

For our proposed OPCR, in the first stage, we propagate observations to missing data, thus we can reduce the time complexity from $\mathcal{O}\left(N^2 + NT^2\right)$ to $\mathcal{O}\left((N^2 + NT^2) \times \rho(1-\rho)\right)$ via sparse attention matrices. Considering the sparsity of the graph structure, it is reasonable to assume that $E$ and $N^2 \times \rho(1-\rho)$ are similar in magnitude. Moreover, since we learn spatial attention matrix from static node features and share it across all spatio-temporal series within a batch, the the computation remains efficient despite the introduction of attention-based confidence assignment. In the second stage, the time complexity of layer-wise updation is $\mathcal{O}\left(ET + NT^2\right)$. As we mentioned

---

[1] https://www.nrel.gov/grid/solar-power-data.html
[2] https://www.ucd.ie/issda/data/commissionforenergyregulationcer/
[3] https://www.iarai.ac.at/traffic4cast/challenge/
[4] https://github.com/Graph-Machine-Learning-Group/spin

in Section 4, after one-step propagation, OPCR requires fewer iterations for refinement compared to SPIN. Therefore, our proposed method actually has greater computational efficiency and scalability.

In addition, following [23], we present the timings and memory consumption in the Table 4. Compared to the spatio-temporal imputation baselines SPIN-H, our proposed model shows significant superiority in computational efficiency and memory usage.

Table 4: Timings and memory consumption.

| MODEL | PV-US | | | CER-E | | |
|---|---|---|---|---|---|---|
| | MEMORY | BATCH/S | BATCH SIZE | MEMORY | BATCH/S | BATCH SIZE |
| BRITS | 11.20GB | 2.84 | 32 | 16.53GB | 1.97 | 32 |
| SAITS | 4.52GB | 8.95 | 128 | 5.28GB | 1.76 | 128 |
| SPIN-H | 14.22GB | 2.88 | 2 | 18.00GB | 3.46 | 2 |
| PRISTI | 14.22GB | 1.40 | 4 | 17.52GB | 1.01 | 4 |
| **OPCR** | 16.98GB | 3.09 | 4 | 18.22GB | 2.87 | 4 |

# B   Proof of Proposition4.3

**Proposition B.1.** *Assume that all ST points are randomly masked with a probability of $\rho$. Let $\mathcal{F}$ be a $K$-iterations imputation model class. If we draw a sample $\tilde{S}$ of size $m$, for any $f \in \mathcal{F}$, the following inequality holds.*

$$\mathbb{P}_{\tilde{S}\sim\tilde{\mathcal{D}}^m}\left[R_{\tilde{\mathcal{D}}}(h_{\tilde{S}}) \leq \epsilon\right] \geq 1 - \delta$$

*with $m \geq \frac{\log 1/\delta}{2\epsilon - 4\eta\mathcal{C}K - 4(1-\eta)\mathcal{C}K \cdot \rho^\tau - 4C_l\Re_m(\mathcal{F})}$.*

Given an sparse training set $\tilde{S} \sim \tilde{\mathcal{D}}^m$, for any $f \in \mathcal{F}$, we have

$$R_{\tilde{\mathcal{D}}}(f) - \hat{R}_{\tilde{S}}(f) = [R_{\tilde{\mathcal{D}}}(f) - R_{\mathcal{D}}(f)] + \left[R_{\mathcal{D}}(f) - \hat{R}_S(f)\right] + \left[\hat{R}_S(f) - \hat{R}_{\tilde{S}}(f)\right] \quad (8)$$

**Corollary B.2.** *For any $f \in \mathcal{F}$ and any $(\boldsymbol{X}, \boldsymbol{Y}) \in \mathcal{X} \times \mathcal{Y}$, we mask all ST points at random with probability of $\rho$. We denote this mask distribution as $\mathcal{M}$. Given a complete spatio-temporal series $\boldsymbol{X}$ and its sparse version $\tilde{\boldsymbol{X}}$, we denote the node-level output of the model as $\hat{\boldsymbol{Y}} = f(\boldsymbol{X})$ and $\tilde{\boldsymbol{Y}} = f(\tilde{\boldsymbol{X}})$. The difference between the model's prediction error on complete data and sparse data have the following upper bound.*

$$\mathbb{E}_{\boldsymbol{M}\sim\mathcal{M}}\left|L(\tilde{\boldsymbol{Y}}, \boldsymbol{Y}) - L(\hat{\boldsymbol{Y}}, \boldsymbol{Y})\right| \leq \mathcal{C} \cdot \sum_{k=1}^{K}\left[\rho^{\tau_{\min}^{k-1}} + \eta \cdot \left(1 - \rho^{\tau_{\max}^{k-1}}\right)\right],$$

*where $\mathcal{C} = C_l \cdot \gamma^K \cdot C_\phi^{K+1} \cdot (d_{\max})^K \cdot B_X B_d$.*

*Proof.* Under assumptions, $f$ is an encoder-decoder architecture, for any complete samples $(\boldsymbol{X}, \boldsymbol{Y})$ and its sparse version $(\tilde{\boldsymbol{X}}, \boldsymbol{Y})$, we have

$$\begin{aligned}\left|L(\tilde{\boldsymbol{Y}}, \boldsymbol{Y}) - L(\hat{\boldsymbol{Y}}, \boldsymbol{Y})\right| &= \left|\frac{1}{N}\sum_{v\in\mathcal{V}}(l(\tilde{\boldsymbol{y}}_v, \boldsymbol{y}_v) - l(\hat{\boldsymbol{y}}_v, \boldsymbol{y}_v))\right| \\ &\leq C_l \cdot \frac{1}{N}\sum_{v\in\mathcal{V}}\|\tilde{\boldsymbol{y}}_v - \hat{\boldsymbol{y}}_v\|_2 \quad (9) \\ &= C_l \cdot \frac{1}{N}\sum_{v\in\mathcal{V}}\left\|f_d(f_e(\tilde{\boldsymbol{X}}))_v - f_d(f_e(\boldsymbol{X}))_v\right\|_2.\end{aligned}$$

Here Eq.9 is due to the loss function $l$ is $C_l$-lipschitz continuous. As the assumption about decoder $f_d$, we have

$$\left| L(\tilde{\boldsymbol{Y}}, \boldsymbol{Y}) - L(\hat{\boldsymbol{Y}}, \boldsymbol{Y}) \right| \leq C_l \cdot \frac{1}{N} \sum_{v \in \mathcal{V}} \left\| f_d(f_e(\tilde{\boldsymbol{X}}))_v - f_d(f_e(\boldsymbol{X}))_v \right\|_2$$

$$= C_l \cdot \frac{1}{N} \sum_{v \in \mathcal{V}} \left\| \phi \left( \frac{1}{T} \sum_{t \in \mathcal{T}} f_e(\tilde{\boldsymbol{X}})_{v,t} \boldsymbol{W}_d \right) - \phi \left( \frac{1}{T} \sum_{t \in \mathcal{T}} f_e(\boldsymbol{X})_{v,t} \boldsymbol{W}_d \right) \right\|_2$$

$$\leq C_l \cdot C_\phi \cdot \frac{1}{NT} \sum_{v \in \mathcal{V}} \left\| \sum_{t \in \mathcal{T}} \left( f_e(\tilde{\boldsymbol{X}})_{v,t} - f_e(\boldsymbol{X})_{v,t} \right) \boldsymbol{W}_d \right\|_2 \tag{10}$$

$$\leq C_l \cdot C_\phi \cdot \frac{1}{NT} \sum_{v \in \mathcal{V}} \sum_{t \in \mathcal{T}} \left\| f_e(\tilde{\boldsymbol{X}})_{v,t} - f_e(\boldsymbol{X})_{v,t} \right\|_2 \cdot \|\boldsymbol{W}_d\|_2$$

$$\leq C_l \cdot C_\phi \cdot B_d \cdot \frac{1}{NT} \sum_{v \in \mathcal{V}} \sum_{t \in \mathcal{T}} \left\| f_e(\tilde{\boldsymbol{X}})_{v,t} - f_e(\boldsymbol{X})_{v,t} \right\|_2. \tag{11}$$

Here Eq.10 is due to the $\phi$ is $C_\phi$-lipschitz continous and Eq.11 is due to $\|\boldsymbol{W}_d\|_2 \leq B_d$. Since the encoder $f_e$ is a $K$-iteration imputation model, we denote the ST point representations after $k$ iterations by $\boldsymbol{H}^k$ and $\tilde{\boldsymbol{H}}^k$. We denote the set of recovered ST points after $k$ iterations by $\mathcal{N}_o^k$. For the number of iterations $K$, we assume that $K$ exactly satisfies $\mathcal{Z}_o^K = \mathcal{Z}$ and for any $\mathcal{Z}_o^{K-1} \subset \mathcal{Z}$. Thus, we have

$$\frac{1}{NT} \sum_{v \in \mathcal{V}} \sum_{t \in \mathcal{T}} \left\| f_e(\tilde{\boldsymbol{X}})_{v,t} - f_e(\boldsymbol{X})_{v,t} \right\|_2 = \frac{1}{NT} \cdot \sum_{v \in \mathcal{V}} \sum_{t \in \mathcal{T}} \left\| \tilde{\boldsymbol{h}}_{v,t}^K - \boldsymbol{h}_{v,t}^K \right\|_2$$

$$= \frac{1}{NT} \cdot \sum_{z \in \mathcal{Z}} \left\| \tilde{\boldsymbol{h}}_z^K - \boldsymbol{h}_z^K \right\|_2, \tag{12}$$

where $\mathcal{Z} = \{(v,t) | v \in \mathcal{V}, t \in \mathcal{T}\}$. Then we have the following recursion:

$$\sum_{z \in \mathcal{Z}_o^k} \left\| \tilde{\boldsymbol{h}}_z^k - \boldsymbol{h}_z^k \right\|_2$$

$$= \sum_{z \in \mathcal{Z}_o^k} \left\| \phi \left( \sum_{z' \in \tilde{\mathcal{N}}_z^k} \tilde{a}_{z' \to z}^k \tilde{\boldsymbol{h}}_{z'}^{k-1} \right) - \phi \left( \sum_{z' \in \mathcal{N}_z} a_{z' \to z} \boldsymbol{h}_{z'}^{k-1} \right) \right\|_2$$

$$\leq C_\phi \cdot \sum_{z \in \mathcal{Z}_o^k} \left\| \sum_{z' \in \tilde{\mathcal{N}}_z^k} \tilde{a}_{z' \to z}^k \tilde{\boldsymbol{h}}_{z'}^{k-1} - \sum_{z' \in \mathcal{N}_z} a_{z' \to z} \boldsymbol{h}_{z'}^{k-1} \right\|_2 \tag{13}$$

$$\leq C_\phi \cdot \sum_{z \in \mathcal{Z}_o^k} \left( \left\| \sum_{z' \in \tilde{\mathcal{N}}_z^k} \tilde{a}_{z' \to z}^k \left( \tilde{\boldsymbol{h}}_{z'}^{k-1} - \boldsymbol{h}_{z'}^{k-1} \right) \right\|_2 + \left\| \sum_{z' \in \mathcal{N}_z} \left( \tilde{a}_{z' \to z}^k - a_{z' \to z} \right) \boldsymbol{h}_{z'}^{k-1} \right\|_2 \right) \tag{14}$$

$$\leq C_\phi \cdot \sum_{z \in \mathcal{Z}_o^k} \left( \sum_{z' \in \tilde{\mathcal{N}}_z^k} |\tilde{a}_{z' \to z}^k| \cdot \left\| \tilde{\boldsymbol{h}}_{z'}^{k-1} - \boldsymbol{h}_{z'}^{k-1} \right\|_2 + \sum_{z' \in \mathcal{N} \backslash \tilde{\mathcal{N}}_z^k} |a_{z' \to z}| \cdot \left\| \boldsymbol{h}_{z'}^{k-1} \right\|_2 + \sum_{z' \in \tilde{\mathcal{N}}_z^k} |\tilde{a}_{z' \to z}^k - a_{z' \to z}| \cdot \left\| \boldsymbol{h}_{z'}^{k-1} \right\|_2 \right)$$

Here Eq.13 is due to that $\phi$ is $C_\phi$-lipschitz continuous. Eq.14 is due to $\tilde{a}_{z' \to z}^k = 0$ if $z' \in \mathcal{N}_z \backslash \tilde{\mathcal{N}}_z^k$.

For $\forall z \in \mathcal{Z}$ and $\forall z' \in \mathcal{N}_z$, we assume that $0 < a_{z' \to z} \leq \gamma$. For $\forall z \in \mathcal{Z}$, $\forall z' \in \mathcal{N}_z^k$ and $\forall k \in [1, K]$, we assume that $0 < \tilde{a}_{z' \to z}^k \leq \gamma$ and $0 < |a_{z' \to z} - \tilde{a}_{z' \to z}^k| \leq \gamma' < \gamma$. These assumptions are easy to implement because the model tends to use positive weights when aggregating neighboring ST points.

For convenience, let $0 < \eta = \frac{\gamma'}{\gamma} < 1$, we have

$$\sum_{z \in \mathcal{Z}_o^k} \left\| \tilde{\bm{h}}_z^k - \bm{h}_z^k \right\|_2$$

$$\leq C_\phi \cdot \sum_{z \in \mathcal{Z}_o^k} \left( \sum_{z' \in \tilde{\mathcal{N}}_z^k} |\tilde{a}_{z' \to z}^k| \cdot \left\| \tilde{\bm{h}}_{z'}^{k-1} - \bm{h}_{z'}^{k-1} \right\|_2 + \sum_{z' \in \mathcal{N} \setminus \tilde{\mathcal{N}}_z^k} |a_{z' \to z}| \cdot \left\| \bm{h}_{z'}^{k-1} \right\|_2 + \sum_{z' \in \tilde{\mathcal{N}}_z^k} |\tilde{a}_{z' \to z}^k - a_{z' \to z}| \cdot \left\| \bm{h}_{z'}^{k-1} \right\|_2 \right)$$

$$\leq C_\phi \cdot \sum_{z \in \mathcal{Z}_o^k} \left( \sum_{z' \in \tilde{\mathcal{N}}_z^k} \gamma \cdot \left\| \tilde{\bm{h}}_{z'}^{k-1} - \bm{h}_{z'}^{k-1} \right\|_2 + \sum_{z' \in \mathcal{N}_z \setminus \tilde{\mathcal{N}}_z^k} \gamma \cdot \left\| \bm{h}_{z'}^{k-1} \right\|_2 + \sum_{z' \in \tilde{\mathcal{N}}_z^k} \gamma' \cdot \left\| \bm{h}_{z'}^{k-1} \right\|_2 \right) \tag{15}$$

$$\leq \gamma C_\phi d_{\max} \cdot \sum_{z \in \mathcal{Z}_o^{k-1}} \left\| \tilde{\bm{h}}_{z'}^{k-1} - \bm{h}_{z'}^{k-1} \right\|_2 + \gamma C_\phi \cdot \sum_{z \in \mathcal{Z}_o^k} \left( \sum_{z' \in \mathcal{N}_z \setminus \tilde{\mathcal{N}}_z^k} \left\| \bm{h}_{z'}^{k-1} \right\|_2 + \sum_{z' \in \tilde{\mathcal{N}}_z} \eta \cdot \left\| \bm{h}_{z'}^{k-1} \right\|_2 \right) \tag{16}$$

For the second term and the third term in Eq. 16, we have

$$\mathbb{E}_{\bm{M}} \left[ \sum_{z \in \mathcal{Z}_o^k} \left( \sum_{z' \in \mathcal{N}_z \setminus \tilde{\mathcal{N}}_z} \left\| \bm{h}_{z'}^{k-1} \right\|_2 + \sum_{z' \in \tilde{\mathcal{N}}_z} \eta \cdot \left\| \bm{h}_{z'}^{k-1} \right\|_2 \right) \right]$$

$$\leq \mathbb{E}_{\bm{M}} \left[ \sum_{z \in \mathcal{Z}} \left( \sum_{z' \in \mathcal{N}_z \setminus \tilde{\mathcal{N}}_z} \left\| \bm{h}_{z'}^{k-1} \right\|_2 + \sum_{z' \in \tilde{\mathcal{N}}_z} \eta \cdot \left\| \bm{h}_{z'}^{k-1} \right\|_2 \right) \right]$$

$$\leq d_{\max} \cdot \left( \rho^{\tau_{\min}^{k-1}} \cdot \sum_{z \in \mathcal{Z}} \left\| \bm{h}_z^{k-1} \right\|_2 + \eta \cdot (1 - \rho^{\tau_{\max}^{k-1}}) \cdot \sum_{z \in \mathcal{Z}} \left\| \bm{h}_z^{k-1} \right\|_2 \right)$$

$$= d_{\max} \cdot \left[ \rho^{\tau_{\min}^{k-1}} + \eta \cdot (1 - \rho^{\tau_{\max}^{k-1}}) \right] \cdot \sum_{z \in \mathcal{Z}} \left\| \bm{h}_z^{k-1} \right\|_2 . \tag{17}$$

Here $\tau_{\max}^k = \max_{z \in \mathcal{Z}} \tau_z^k$ and $\tau_{\min}^k = \min_{z \in \mathcal{Z}} \tau_z^k$, where $\tau_z^k = |\{z' | d_{z' \to z} \leq k, z' \in \mathcal{Z}_o^{k-1}\}|$. Actually, $\tau_z^k$ represents the number of $k$-hop neighbors of ST point $z$.

For $\sum_{z \in \mathcal{Z}} \left\| \bm{h}_z^k \right\|_2$, we have

$$\sum_{z \in \mathcal{Z}} \left\| \bm{h}_z^k \right\|_2 = \sum_{z \in \mathcal{Z}} \left\| \phi \left( \sum_{z' \in \mathcal{N}_z} a_{z' \to z} \bm{h}_{z'}^{k-1} \right) \right\|_2$$

$$= \sum_{z \in \mathcal{Z}} \left\| \phi \left( \sum_{z' \in \mathcal{N}_z} a_{z' \to z} \bm{h}_{z'}^{k-1} \right) - \phi(\bm{0}) \right\|_2$$

$$\leq C_\phi \cdot \sum_{z \in \mathcal{Z}} \left\| \sum_{z' \in \mathcal{N}_z} a_{z' \to z} \bm{h}_{z'}^{k-1} \right\|_2$$

$$\leq \gamma C_\phi \cdot \sum_{z \in \mathcal{Z}} \sum_{z' \in \mathcal{N}_z} \left\| \bm{h}_{z'}^{k-1} \right\|_2$$

$$\leq \gamma C_\phi d_{\max} \cdot \sum_{z \in \mathcal{Z}} \left\| \bm{h}_z^{k-1} \right\|_2$$

$$\leq (\gamma C_\phi d_{\max})^k \cdot \sum_{z \in \mathcal{Z}} \left\| \bm{h}_z^0 \right\|_2$$

$$\leq (\gamma C_\phi d_{\max})^k \cdot NTB_X . \tag{18}$$

Therefore, expanding the recursion in Eq. 16, we have

$$
\mathbb{E}_{\boldsymbol{M}}\left[\sum_{z\in\mathcal{Z}_o^k}\left\|\tilde{\boldsymbol{h}}_z^k-\boldsymbol{h}_z^k\right\|_2\right]
$$

$$
\leq \mathbb{E}_{\boldsymbol{M}}\left[\gamma C_\phi d_{\max}\cdot\sum_{z\in\mathcal{Z}_o^{k-1}}\left\|\tilde{\boldsymbol{h}}_{z'}^{k-1}-\boldsymbol{h}_{z'}^{k-1}\right\|_2+\gamma C_\phi\cdot\sum_{z\in\mathcal{Z}_o^k}\left(\sum_{z'\in\mathcal{N}_z\setminus\tilde{\mathcal{N}}_z^k}\left\|\boldsymbol{h}_{z'}^{k-1}\right\|_2+\sum_{z'\in\tilde{\mathcal{N}}_z}\eta\cdot\left\|\boldsymbol{h}_{z'}^{k-1}\right\|_2\right)\right]
$$

$$
\leq \gamma C_\phi d_{\max}\cdot\left(\mathbb{E}_{\boldsymbol{M}}\left[\sum_{z\in\mathcal{Z}_o^{k-1}}\left\|\tilde{\boldsymbol{h}}_{z'}^{k-1}-\boldsymbol{h}_{z'}^{k-1}\right\|_2\right]+\left[\rho^{\tau_{\min}^{k-1}}+\eta\cdot(1-\rho^{\tau_{\max}^{k-1}})\right]\cdot\sum_{z\in\mathcal{Z}}\left\|\boldsymbol{h}_z^{k-1}\right\|_2\right)
$$

$$
\leq \gamma C_\phi d_{\max}\cdot\mathbb{E}_{\boldsymbol{M}}\left[\sum_{z\in\mathcal{Z}_o^{k-1}}\left\|\tilde{\boldsymbol{h}}_{z'}^{k-1}-\boldsymbol{h}_{z'}^{k-1}\right\|_2\right]+\left[\rho^{\tau_{\min}^{k-1}}+\eta\cdot(1-\rho^{\tau_{\max}^{k-1}})\right]\cdot(\gamma C_\phi d_{\max})^k\cdot NTB_X.
$$

$$(19)$$

Then we can derive

$$
\mathbb{E}_{\boldsymbol{M}}\left[\sum_{z\in\mathcal{Z}}\left\|\tilde{\boldsymbol{h}}_z^K-\boldsymbol{h}_z^k\right\|_2\right]=\mathbb{E}_{\boldsymbol{M}}\left[\sum_{z\in\mathcal{Z}_o^K}\left\|\tilde{\boldsymbol{h}}_z^K-\boldsymbol{h}_z^k\right\|_2\right]
$$

$$
\leq \gamma C_\phi d_{\max}\cdot\mathbb{E}_{\boldsymbol{M}}\left[\sum_{z\in\mathcal{Z}_o^{k-1}}\left\|\tilde{\boldsymbol{h}}_{z'}^{k-1}-\boldsymbol{h}_{z'}^{k-1}\right\|_2\right]+\left[\rho^{\tau_{\min}^{k-1}}+\eta\cdot(1-\rho^{\tau_{\max}^{k-1}})\right]\cdot(\gamma C_\phi d_{\max})^k\cdot NTB_X
$$

$$
\leq (\gamma C_\phi d_{\max})^K\cdot NTB_X\cdot\sum_{k=1}^{K}\left[\rho^{\tau_{\min}^{k-1}}+\eta\cdot(1-\rho^{\tau_{\max}^{k-1}})\right].\tag{20}
$$

Substituting Eq.20 into Eq.12, we have

$$
\mathbb{E}_{\boldsymbol{M}}\left|L(\tilde{\boldsymbol{Y}},\boldsymbol{Y})-L(\hat{\boldsymbol{Y}},\boldsymbol{Y})\right|\leq C_l\cdot C_\phi\cdot B_d\cdot\frac{1}{NT}\cdot\mathbb{E}_{\boldsymbol{M}}\left[\sum_{z\in\mathcal{Z}}\left\|\tilde{\boldsymbol{h}}_z^K-\boldsymbol{h}_z^K\right\|_2\right]
$$

$$
\leq C_l\cdot C_\phi\cdot B_d\cdot\frac{1}{NT}\cdot(\gamma C_\phi d_{\max})^K\cdot NTB_X\cdot\sum_{k=1}^{K}\left[\rho^{\tau_{\min}^{k-1}}+\eta\cdot(1-\rho^{\tau_{\max}^{k-1}})\right]
$$

$$
\leq C_l\cdot\gamma^K\cdot C_\phi^{K+1}\cdot(d_{\max})^K\cdot B_X B_d\cdot\sum_{k=1}^{K}\left[\rho^{\tau_{\min}^{k-1}}+\eta\cdot(1-\rho^{\tau_{\max}^{k-1}})\right].
$$

Let $\mathcal{C}=C_l\cdot\gamma^K\cdot C_\phi^{K+1}\cdot(d_{\max})^K\cdot B_X B_d$, we finally derive the follows inequality.

$$
\mathbb{E}_{\boldsymbol{M}}\left|L(\tilde{\boldsymbol{Y}},\boldsymbol{Y})-L(\hat{\boldsymbol{Y}},\boldsymbol{Y})\right|\leq\mathcal{C}\cdot\sum_{k=1}^{K}\left[\rho^{\tau_{\min}^{k-1}}+\eta\cdot(1-\rho^{\tau_{\max}^{k-1}})\right].
$$

$\square$

For the second term in Eq.8, $R_{\mathcal{D}}(f)-\hat{R}_S(f)$ represents the generalization performance of $f\in\mathcal{F}$ over complete dataset. Model complexity measurements, such as VC-Dimension [27], Rademacher complexity [28] and Covering number [29] are related to the generalization ability. Therefore, they can provide useful information about number of training samples $m$ required for PAC-learnability under complete data. Rademacher complexity is frequently applied to investigate the generalization performances of GNN-based models. The Rademacher complexity-based generalization bound as given in [15] is:

**Lemma B.3.** *Let $\mathcal{H}$ be a function class mapping from $\mathcal{X}$ to $[0,1]$. Then, for any $\delta>0$, with probability at least $1-\delta$ over the draw of an i.i.d. sample $S$ of size $m$, for any $h\in\mathcal{H}$, the following*

*inequality holds.*

$$R_{\mathcal{D}}(h) \leq \hat{R}_S(h) + 2\Re_m(\mathcal{H}) + \sqrt{\frac{\log 1\backslash\delta}{2m}}$$

As our assumptions, the loss function $l : \mathbb{R} \times \mathcal{Y} \to [0,1]$ is $C_l$-lipschitz continuous. Then we can derive the following corollary, which show that generalization gap is small when provided large enough training samples.

**Corollary B.4.** *Let $\mathcal{F} : \mathcal{X} \to [0,1]$ be a function class for spatio-temporal data imputation. function class mapping from $\mathcal{X}$ to $[0,1]$. Then, for any $\delta > 0$, with probability at least $1 - \delta$ over the draw of an i.i.d. sample $S$ of size $m$, for any $h \in \mathcal{H}$, the following inequality holds.*

$$P\left(\left|R_{\mathcal{D}}(f) - \hat{R}_S(f)\right| \geq \epsilon\right) \leq e^{-2m(\epsilon - 2C_l\Re_m(\mathcal{F}))}.$$

*Proof.* Using Lemma B.3 and contraction lemma[15], we have

$$R_{\mathcal{D}}(f) \leq \hat{R}_S(f) + 2\Re_m(l \circ \mathcal{F}) + \sqrt{\frac{\log 1\backslash\delta}{2m}}$$

$$\leq \hat{R}_S(f) + 2C_l\Re_m(\mathcal{F}) + \sqrt{\frac{\log 1\backslash\delta}{2m}}.$$

Therefore, we can rewrite this inequality then we have for any $f \in \mathcal{F}$

$$P\left(\left|R_{\mathcal{D}}(f) - \hat{R}_S(f)\right| \geq \epsilon\right) \leq e^{-2m(\epsilon - 2C_l\Re_m(\mathcal{F}))}.$$

$\square$

Using Corollary B.2 and B.4, we have

$$P\left(\left|R_{\tilde{\mathcal{D}}}(f) - \hat{R}_{\tilde{S}}(f)\right| \geq \epsilon\right)$$

$$\leq P\left(\left|R_{\mathcal{D}}(f) - \hat{R}_S(f)\right| \geq \epsilon - \left|R_{\tilde{\mathcal{D}}}(f) - R_{\mathcal{D}}(f) + \hat{R}_S(f) - \hat{R}_{\tilde{S}}(f)\right|\right)$$

$$\leq P\left(\left|R_{\mathcal{D}}(f) - \hat{R}_S(f)\right| \geq \epsilon - 2\mathcal{C} \cdot \sum_{k=1}^{K}\left[\rho^{\tau_{\min}^{k-1}} + \eta \cdot (1 - \rho^{\tau_{\max}^{k-1}})\right]\right)$$

$$\leq \exp\left(-2m\left[\epsilon - 2\mathcal{C} \cdot \sum_{k=1}^{K}\left(\eta - \eta\rho^{\tau_{\max}^{k-1}} + \rho^{\tau_{\min}^{k-1}}\right) - 2C_l\Re_m(\mathcal{F})\right]\right).$$

Then we derive the number of required samples $m$:

$$\exp\left(-2m\left[\epsilon - 2\mathcal{C} \cdot \sum_{k=1}^{K}\left(\eta - \eta\rho^{\tau_{\max}^{k-1}} + \rho^{\tau_{\min}^{k-1}}\right) - 2C_l\Re_m(\mathcal{F})\right]\right) \leq \delta$$

$$\Rightarrow \exp\left(2m\left[\epsilon - 2\mathcal{C} \cdot \sum_{k=1}^{K}\left(\eta - \eta\rho^{\tau_{\max}^{k-1}} + \rho^{\tau_{\min}^{k-1}}\right) - 2C_l\Re_m(\mathcal{F})\right]\right) \geq \frac{1}{\delta}$$

$$\Rightarrow 2m\left[\epsilon - 2\mathcal{C} \cdot \sum_{k=1}^{K}\left(\eta - \eta\rho^{\tau_{\max}^{k-1}} + \rho^{\tau_{\min}^{k-1}}\right) - 2C_l\Re_m(\mathcal{F})\right] \geq \log(\frac{1}{\delta})$$

$$\Rightarrow m \geq \frac{1}{2\epsilon - 4\mathcal{C} \cdot \sum_{k=1}^{K}\left(\eta - \eta\rho^{\tau_{\max}^{k-1}} + \rho^{\tau_{\min}^{k-1}}\right) - 4C_l\Re_m(\mathcal{F})} \cdot \log\sqrt{\frac{1}{\delta}}$$

$$\Rightarrow m \geq \frac{1}{2\epsilon - 4\eta\mathcal{C}K - 4(1-\eta)\mathcal{C} \cdot \sum_{k=1}^{K}\rho^{\tau_{\max}^{k-1}} - 4C_l\Re_m(\mathcal{F})} \cdot \log\sqrt{\frac{1}{\delta}}$$

$$\Rightarrow m \geq \frac{1}{2\epsilon - 4\eta\mathcal{C}K - 4(1-\eta)\mathcal{C}K \cdot \rho^{\tau_{\max}^{K-1}} - 4C_l\Re_m(\mathcal{F})} \cdot \log\sqrt{\frac{1}{\delta}}$$

For convenience, we let $\tau = \tau_{\max}^{K-1}$.

# C   Proof of Proposition 5.1

*Proof.* Similar to proof of Proposition 5.1, we first derive the following upper bound for the difference between model performances under complete and sparse data. For any $M \in \mathcal{M}$, we have

$$\frac{1}{NT} \cdot \sum_{z \in \mathcal{Z}} \left\| \tilde{\boldsymbol{h}}_z - \boldsymbol{h}_z \right\|_2$$

$$= \frac{1}{NT} \cdot \sum_{z \in \mathcal{Z}} \| \sigma(\tilde{\boldsymbol{a}}_z) \boldsymbol{V} - \sigma(\boldsymbol{a}_z) \boldsymbol{V} \|_2$$

$$\leq \frac{1}{NT} \cdot \sum_{z \in \mathcal{Z}} \left\| \sum_{z' \in \mathcal{Z}_o} \frac{a_{z' \to z}}{\sum_{z'' \in \mathcal{Z}_o} a_{z'' \to z}} \boldsymbol{v}_{z'} - \sum_{z' \in \mathcal{Z}} \frac{a_{z' \to z}}{\sum_{z'' \in \mathcal{Z}} a_{z'' \to z}} \boldsymbol{v}_{z'} \right\|_2$$

$$\leq \frac{B_v}{NT} \cdot \sum_{z \in \mathcal{Z}} \left| \sum_{z' \in \mathcal{Z}_o} \left( \frac{a_{z' \to z}}{\sum_{z'' \in \mathcal{Z}_o} a_{z'' \to z}} - \frac{a_{z' \to z}}{\sum_{z'' \in \mathcal{Z}} a_{z'' \to z}} \right) \right| + \frac{1}{NT} \cdot \sum_{z \in \mathcal{Z}} \left| \sum_{z' \in \mathcal{Z} \setminus \mathcal{Z}_o} \frac{a_{z' \to z}}{\sum_{z'' \in \mathcal{Z}} a_{z'' \to z}} \right|,$$

where $\max_{z \in \mathcal{Z}} \| \boldsymbol{v}_z \|_2 \leq B_v$. Then we have

$$\mathbb{E}_M \left[ \frac{1}{NT} \cdot \sum_{z \in \mathcal{Z}} \left\| \tilde{\boldsymbol{h}}_z - \boldsymbol{h}_z \right\|_2 \right]$$

$$\leq \frac{B_v}{NT} \cdot \sum_{z \in \mathcal{Z}} \mathbb{E}_M \left[ \left| \sum_{z' \in \mathcal{Z}_o} \frac{a_{z' \to z} \cdot \left( \sum_{z'' \in \mathcal{Z} \setminus \mathcal{Z}_o} a_{z'' \to z} \right)}{\left( \sum_{z'' \in \mathcal{Z}_o} a_{z'' \to z} \right) \cdot \left( \sum_{z'' \in \mathcal{Z}} a_{z'' \to z} \right)} \right| + \left| \sum_{z' \in \mathcal{Z} \setminus \mathcal{Z}_o} \frac{a_{z' \to z}}{\sum_{z'' \in \mathcal{Z}} a_{z'' \to z}} \right| \right]$$

$$= \frac{2 B_v}{NT} \cdot \sum_{z \in \mathcal{Z}} \mathbb{E}_M \left| \frac{\sum_{z' \in \mathcal{Z} \setminus \mathcal{Z}_o} a_{z' \to z}}{\sum_{z' \in \mathcal{Z}} a_{z' \to z}} \right|$$

$$\leq 2 B_v \cdot \rho$$

Similar to proof of Proposition 4.3,

$$P \left( \left| R_{\tilde{\mathcal{D}}}(f) - \hat{R}_{\tilde{S}}(f) \right| \geq \epsilon \right) \leq \exp \left( -2m \left[ \epsilon - 4 C_l \cdot C_\phi \cdot B_d \cdot B_v \cdot \rho - 2 C_l \Re_m(\mathcal{F}) \right] \right)$$

$$\Rightarrow P \left( \left| R_{\tilde{\mathcal{D}}}(f) - \hat{R}_{\tilde{S}}(f) \right| \geq \epsilon \right) \leq \exp \left( -2m \left[ \epsilon - 4 \mathcal{C} \cdot \rho - 2 C_l \Re_m(\mathcal{F}) \right] \right)$$

$$\Rightarrow \exp \left( -2m \left[ \epsilon - 4 \mathcal{C} \cdot \rho - 2 C_l \Re_m(\mathcal{F}) \right] \right) \leq \delta$$

$$\Rightarrow 2m \left[ \epsilon - 4 \mathcal{C} \cdot \rho - 2 C_l \Re_m(\mathcal{F}) \right] \geq \log \frac{1}{\delta}$$

$$\Rightarrow m \geq \frac{\log 1/\delta}{2 \left[ \epsilon - 4 \mathcal{C} \cdot \rho - 2 C_l \Re_m(\mathcal{F}) \right]},$$

where $\mathcal{C} = C_l \cdot C_\phi \cdot B_d \cdot B_v$. $\qquad\square$

