# OpenReview forum: "Learning from Highly Sparse Spatio-temporal Data"
_NeurIPS.cc/2024/Conference — NeurIPS 2024 poster_

### Official Review · Reviewer_aMi7 · 2024-07-01

**Soundness:** 3
**Presentation:** 1
**Contribution:** 2
**Rating:** 5
**Confidence:** 3

**Summary:**

This paper proposes to address the challenge of learning from incomplete spatio-temporal data, which is prevalent in various real-world applications. Accordingly, this paper proposes a method named OPCR to handle data sparsity more effectively. Specifically, the method first directly utilizes spatial and temporal relationships to propagate limited observations to the global context through a sparse attention mechanism. After the initial propagation, confidence-based refinement is applied to refine the imputations, correlating the missing data with valid data to enhance the quality of the imputation. Experiments show that OPCR outperforms several baselines in various downstream tasks involving highly sparse spatio-temporal data.

**Strengths:**

S1. Tackling spatio-temporal data with high sparsity is an important topic. It is critical to design effective method to tackle such a case, which has many downstream applications.

S2. The paper offers a theoretical analysis on the advantages of the proposed method against existing models. The design proposed method is based on the derived theory.

S3. Experiments show that the proposed model achieves the best performance among several baselines in different settings and various spatio-temporal tasks.

**Weaknesses:**

W1. The presentation needs to be significantly improved. Many parts of the contents are vague and difficult to follow.

-The introduction section presents most of the contents that should be separately discussed in the related work section, while the related work section is missing.

-The notations for the theory provided in Section 3 do not offer detailed explanations. I cannot find the definitions for many of them. For example, what is “poly()” in Definition 3.1? what is \mathcal{T}? what are Z_m, Z_o?

-In section 4.1.1, the update in GNN equation uses S_l, but then h_^{s}_{v,t} is used to denote the equation. Are they the same thing? In section 4.1.2, what is \hat{X}, which does not appear anywhere previously?

I don’t really follow the paper, as so many of the notations are not properly defined and explained.

W2. It is not clear to me why the proposed belongs to “one-step propagation”.  Stacking GNN layers implicitly means several iterations of propagation, which is how section 4.1.1 presents. Besides, the temporal self-attention has nothing to do with “propagation”.

W3. It is not clear to me how sparse spatial attention works. The GNN equation and Figure 1 indicates that the original graph structure is utilized. However, in equation 4, it is mentioned only “available nodes” are selected to perform self-attention. In this case, only the representations of nodes that have observed data are updated? How exactly this module functions is not clear.

W4. Figure 1 is not informative at all. I cannot really understand which parts of the nodes and timestamps are utilized for sparse attention. Again, due to bad notations and presentation, I don’t really follow the paper.

W5. Some of the related studies are highly related, and thus should be included. For example, [1, 2].

[1] Handling Missing Data with Graph Representation Learning. NeurIPS 2020

[2] Networked Time Series Imputation via Position-aware Graph Enhanced Variational Autoencoders. KDD 2023

**Questions:**

Please give more explanations on my comments in W1-W4.

**Limitations:**

The authors did not discuss the limitations or negative impact of their paper.

---

> ### Author Rebuttal · Authors · 2024-08-07
>
> Thanks for your careful reviews and insightful suggestions. We greatly appreciate your feedback. Please see the below responses to your comments (see global response to your questions about more baselines and confused notations).
> > **[Weakness 1.1]**  The introduction section presents most of the contents that should be separately discussed in the related work section, while the related work section is missing.
>
> Thanks for your suggestions. In the revised version, we have reorganized the paper to make it clearer. Specifically, we have streamlined the introduction to focus only on the background and significance of the spatio-temporal data imputation, briefly analyze the limitations of existing methods, and finally overview of our work. In addition, we have discussed related work in a separate section, including graph imputation, time-series imputation, and spatio-temporal data imputation, as well as comparing our method with them to highlight our advancements and contributions.
>
> > **[Weakness 1.3.2]** In section 4.1.2, what is \hat{X}, which does not appear anywhere previously? I don’t really follow the paper, as so many of the notations are not properly defined and explained.
>
> We appreciate the constructive feedback. For the sparse temporal attention module in Sec. 4.1.2, we consider timestamp encoding and positional encoding to introduce sequence information. Specifically, the input $\bar X$ can be formulated as:
> $$
> \bar{X} = X + PE(X) + MLP(U),
> $$
> where $U$ is the available real-world time information, such as the hour of the day; $PE$ is the vanilla positional encoding as follows.
> $$
> PE_{(pos, 2i)} = sin(pos/10000^{2i/d_{model}}) ,
> $$
>
> $$
> PE_{(pos, 2i+1)} = cos(pos/10000^{2i/d_{model}}).
> $$
>
>
> > **[Weakness 1.3.1]** In section 4.1.1, the update in GNN equation uses S_l, but then h_^{s}_{v,t} is used to denote the equation. Are they the same thing?
> > **[Weakness 2]** It is not clear to me why the proposed belongs to “one-step propagation”. Stacking GNN layers implicitly means several iterations of propagation, which is how section 4.1.1 presents. Besides, the temporal self-attention has nothing to do with “propagation”.
> > **[Weakness 3]** It is not clear to me how sparse spatial attention works. The GNN equation and Figure 1 indicates that the original graph structure is utilized. However, in equation 4, it is mentioned only “available nodes” are selected to perform self-attention. In this case, only the representations of nodes that have observed data are updated? How exactly this module functions is not clear.
>
> Thanks for your comments. The sparse spatial attention module may not be expressed clearly enough. Let us first clarify this module and then answer your questions one by one.
>
> - **Using multi-layers GNN:** In Lines 150 to 155, $L_s$-layers GCN was utilized to learn static node spatial features from topology structure, and the learned static spatial features are the output $S_{L_s}$. Formally, this part does not belong to the sparse spatial attention that is dedicated to capturing dynamic spatial information. To avoid confusion, we will add a new section before Sec. 4.1.1, titled Learning of Static Spatial and Temporal Features, and move this part and our response to your [Weakness 1.3.2] to this section.
> - **The core of sparse spatial attention module:** The sparse spatial attention module captures the comprehensive correlations of nodes based on the static node spatial features learned by $L_s$-layer GCN, i.e., the input node embedding $S$ of this module is $S_{L_s}$. Then, $S$, $S$, and $X_t$ serve as query, key, and value, respectively, to learn the spatial dependencies-based representation ${h}^{s}_{v,t}$ of each ST point $(v,t)$ that lacks features.
> - **The object of attention mechanism:** We need to obtain representation of the missing point $(v, t)$, while the observed point keeps the original feature unchanged. Then, according to Eq. 5, the attention mechanism acts between the missing points and the observed points, and finally, the weighted observed point representations are used to update the missing point representation.
>
> Based on this, we address each of your questions as follows.
>
> ***Q1: Are $S_l$ and $h^s_{v,t}$ the same thing?***
>
> $S_l$ and $h^s_{v,t}$ are not equivalent. $S_l$ denotes the learned static node spatial feature by $L_s$-layers GCN. ${h}^{s}_{v,t}$ denotes the learned dynamic representation by spatial attention module for ST point $(v,t)$.
>
> ***Q2: Why the proposed belongs to “one-step propagation”.***
>
> In dynamic spatio-temporal data learning, we focus on the propagation at each time step, i.e., the steps of dynamic propagation. The node spatial features learned by multi-layer GCN are static and shared across all time steps, which are not counted in the dynamic propagation. The dynamic process is reflected in the two proposed attention modules; each unobserved node receives information from all observations once, which is achieved by one matrix multiplication, so we refer this strategy as “one-step propagation”.
>
> ***Q3: Only the representations of nodes that have observed data are updated?***
>
> Only the representations of the missing data are updated by aggregating the representation of all observed data based on the attention weights between the missing data and observed data. As for the observed data, their representations use the original feature representation and are not updated.
>
> > **[Weakness 4]** Figure 1 is not informative at all. I cannot really understand which parts of the nodes and timestamps are utilized for sparse attention.
>
> Thanks for your valuable feedback. We have updated Figure 1 to include additional annotations, as shown in Fig. 1 in the rebuttal-submitted PDF. Specifically, for the sparse spatial attention module, we first learn the sparse attention matrix from structural embeddings $S$ using all observed nodes. At each time step $t$, we propagate observed nodes' information to missing data with attention-based weight.

---

> > ### Comment · Reviewer_aMi7 · 2024-08-13
> >
> > Thanks for your response. After reading it, I would like to keep my score.

---

> > > ### Author Response · Authors · 2024-08-13
> > > **Welcome for more discussions**
> > >
> > > Thanks for your valuable time in reviewing and constructive comments, according to which we have tried our best to answer the questions and carefully revise the paper. We humbly hope our response has addressed your concerns.
> > >
> > > Considering your current rating, if you believe that our responses have satisfactorily addressed your concerns, we kindly request that you consider revising your final rating of our manuscript. If you have any additional concerns or comments that we may have missed in our responses, we would be most grateful for any further feedback from you to help us further enhance our work.

---

> > > > ### Comment · Area_Chair_sNGY · 2024-08-14
> > > > **need your comment**
> > > >
> > > > To reviewer aMi7,
> > > >
> > > > In response to the authors' comments, it is unclear in what respect they remain problematic.
> > > > If you are going to keep the initial score, please explain in detail why.
> > > >
> > > > best
> > > >
> > > > Area Chair,

---

### Official Review · Reviewer_g4JR · 2024-07-09

**Soundness:** 3
**Presentation:** 4
**Contribution:** 3
**Rating:** 7
**Confidence:** 4

**Summary:**

The paper addresses the issue of incomplete spatio-temporal data. It theoretically analyzes how existing iterative message-passing methods are susceptible to the impacts of data sparsity and graph sparsity. It proposes the One-step Propagation and Confidence-based Refinement (OPCR). In OPCR, the Sparse Spatial Attention module captures spatial information, while the Sparse Temporal Attention module focuses on dynamic temporal information. And, the Confidence Module in Confidence-based Iterative Refinement further refines these two types of information through spatio-temporal dependencies.

**Strengths:**

1.	The paper is well-organized, identifying limitations with existing methods through theoretical analysis and thereby motivating the proposal of an improved method for modeling spatio-temporal information.

2.	The theoretical analysis of existing iterative message-passing methods is interesting and may inspire future research in the sparse data learning.

3.	A good one-step propagation strategy that can avoid error accumulation and provide a more efficient learning process.

**Weaknesses:**

1.	The paper uses PAC-learnability to analyze generalization risk. Have the author(s) considered using other mathematical tools, such as Rademacher complexity?
2.	Could the author(s) further discuss the potential strength of the proposed method in practical scenarios?
3.	There are some typo and grammatical errors. For instance, "seperate" on Line 13, "problem" on Line 29.

**Questions:**

1.	Have the author(s) considered using other mathematical tools to analyze generalization risk, such as Rademacher complexity?
2.	Could the author(s) further discuss the potential strength of the proposed method in practical scenarios?

**Limitations:**

yes

---

> ### Author Rebuttal · Authors · 2024-08-07
>
> Many thanks for your positive comments and constructive feedback. Please see the below responses to your comments.
> > **[Weakness 1 & Question 1]** The paper uses PAC-learnability to analyze generalization risk. Have the author(s) considered using other mathematical tools, such as Rademacher complexity?
> >
>
> Thanks for this valuable feedback. Other tools, such as Rademacher complexity, are usually closely related to the model structure, and are therefore model-specific mathematical tools. However, our work aims to investigate the impact of missing data on the learning ability of (general) models and to seek breakthroughs in theoretically inspired methods based on these insights, which can be facilitated by the PAC-Learnability theory rather than model-specific theory. The motivation for this theoretical analysis is that the influencing factors caused by missing data, as revealed through the general models, are universal and not confined to any specific model. This broad applicability can inspire improvements to various models (rather than specific models). It is undeniable that studying the generalization error (or convergence rate) of specific models is also of significant importance and will be the direction of our future work.
>
> > **[Weakness 2 & Question 2]** Could the author(s) further discuss the potential strength of the proposed method in practical scenarios?
>
>
> Thank you for your question. Our proposed method has three potential strengths:
>
> - **Lowering the data threshold:** Our method addresses both device failure (point missing) and device unavailability (spatial missing) in spatio-temporal data imputation. Learning from spatial missing data allows for the generalization of local observations to global data, which can reduce research costs in various fields.
> - **Correlation mining:** Our approach provides inherent spatial and temporal context for message-passing rather than just propagating information along the spatiotemporal structure. This allows for easier encapsulation of static spatio-temporal information.
> - **Parallel recovery:** Our one-step propagation aggregates all observations to target data, enabling parallel processing of missing data in practical scenarios. This avoids the low computational efficiency of iterative imputation methods, which makes it particularly suitable for large-scale datasets.
>
> > **[Weakness 3]** There are some typo and grammatical errors. For instance, "seperate" on Line 13, "problem" on Line 29.
> >
>
> Thanks for your careful review. We have modified and checked these typos and grammatical errors in the revised version.

---

> > ### Comment · Reviewer_g4JR · 2024-08-13
> >
> > Thank you for the detailed responses, it addressed most of my concerns.

---

### Official Review · Reviewer_fc8s · 2024-07-12

**Soundness:** 3
**Presentation:** 3
**Contribution:** 3
**Rating:** 8
**Confidence:** 4

**Summary:**

This paper proposes a sparse attention-based one-step imputation and confidence-based refinement approach named One-step Propagation and Confidence-based Refinement (OPCR). The authors evaluate the proposed model across two downstream tasks involving highly sparse spatio-temporal data. The contributions of this paper are as follows:

1. The authors provide a theoretical analysis of a general spatio-temporal iterative imputation model from the perspective of PAC-learnability.

2. Motivated by the theoretical results, this paper introduces a sparse attention-based one-step propagation strategy. This strategy directly propagates information from limited observations to all missing data by leveraging inherent spatial and temporal relationships, resulting in two separate spatial and temporal imputation results. The authors then perform confidence-based spatio-temporal refinement to eliminate the bias introduced by the separate imputations by assigning confidence-based propagation weights to the imputation results.

3. Finally, experiments are conducted comparing several existing imputation methods with the proposed method on real-world datasets, demonstrating the effectiveness of the proposed method.

**Strengths:**

1. The theoretical analysis of the general spatio-temporal iterative imputation model from the PAC-learnability perspective provided in this paper is relatively innovative, explaining the error accumulation caused by multiple iterations. The authors also present a PAC-learnability analysis for sparse attention-based imputation models and provide detailed proofs for both theoretical analyses in the appendix.

2. This paper proposes a novel one-step propagation and confidence-based refinement framework. This framework addresses the problems of error accusation and can be applied to spatial and temporal missing scenes. The high-level idea may provide novel insight for further imputation algorithm design.

3. This paper explores spatially sparse data in the experiments, which helps to make full use of available data and lowers the barriers for implementing spatio-temporal models in real scenarios.

**Weaknesses:**

1. In Section 4.1.2, the construction details of the input to the temporal sparse attention module are not described. Additionally, there are some typos in this paper. For example, the symbol \(H^0\) at line 196 is incorrectly written, the symbols $q^t_t$ and $k^t_k$ in Equation 7 are irregular and inconsistent with the corresponding symbols in Equation 6, and some symbols in the two formulas under line 195 lack descriptions.

2. Some diffusion model-based methods, such as CSDI [1] and PriSTI [2], can propagate information from observed data to missing data directly without multiple iterations. It would be better to clarify the advantages of the proposed model compared to these methods.

References:

[1] Tashiro Y, Song J, Song Y, et al. Csdi: Conditional score-based diffusion models for probabilistic time series imputation[J]. Advances in Neural Information Processing Systems, 2021, 34: 24804-24816.

[2] Liu M, Huang H, Feng H, et al. Pristi: A conditional diffusion framework for spatiotemporal imputation[C]//2023 IEEE 39th International Conference on Data Engineering (ICDE). IEEE, 2023: 1927-1939.

**Questions:**

1. In Figures 4 and 5, why is the MAE result of the SAITS ∞ in the PV-US and CER-E datasets?

2. What's the advantage of the proposed model compared with diffusion model-based methods?

**Limitations:**

Yes.

---

> ### Author Rebuttal · Authors · 2024-08-07
>
> Many thanks for your positive comments and constructive feedback. Please see the below responses to your comments.
> > **[Weakness 1.1]** In Section 4.1.2, the construction details of the input to the temporal sparse attention module are not described.
>
> Thanks for your question. For the temporal sparse attention module, we consider timestamp encoding and positional encoding to introduce sequence information. Specifically, for any node $v \in \mathcal V$, the input $\bar X_v$ can be formulated as:
> $$
> \bar{X_v} = X_v + PE(X_v) + MLP(U),
> $$
> where $X_v$ is the associated time-series of node $v$; $U$ is the available real-world time information, such as the hour of the day; $PE$ is the vanilla positional encoding as follows.
> $$
> PE_{(pos, 2i)} = sin(pos/10000^{2i/d_{model}}),
> $$
>
> $$
> PE_{(pos, 2i+1)} = cos(pos/10000^{2i/d_{model}}).
> $$
>
>
> > **[Weakness 1.2]** Additionally, there are some typos in this paper. For example, the symbol (H^0) at line 196 is incorrectly written, the symbols 𝑞𝑡𝑡 and 𝑘𝑘𝑡 in Equation 7 are irregular and inconsistent with the corresponding symbols in Equation 6, and some symbols in the two formulas under line 195 lack descriptions.
>
> Thanks for your careful review. We have corrected these typos and thoroughly checked the symbols in the revised version.
> Specifically, in Section 4.2, to avoid confusion, we replace "$H$" with "$O$" to denote the learned representations in the second stage. Then, the layer-wise updation can be formulated as
> $$
> O_{v,t}^{l+1} = \text{MLP}  \left( O_{v,t}^{l} || \sum_{(v',t')\in {N_{v,t}}}  \beta_{v't'} \cdot O_{v',t'}^l  \right),
> $$
> where $O^0 = h_{v,t}^s + {h}_{v,t}^t$.
>
> For the sparse attention-based confidence, we have revised inconsistent symbols and rewritten Eq.7 as follows.
> $$
> \beta_{vt} =  \frac{\sum_{k\in \tilde{V}}\exp (<q_v^s, k_k^s>)}{\sum_{k\in V}\exp (<q_v^s, k_k^s>)} +  \frac{\sum_{k\in \tilde{T_v}}\exp (<q_{v,t}^t, k_{v,k}^t>)}{\sum_{k\in T}\exp (<q_{v,t}^t, k_{v,k}^t>)}
> $$
>
>
> > **[Weakness 2]** Some diffusion model-based methods, such as CSDI [1] and PriSTI [2], can propagate information from observed data to missing data directly without multiple iterations. It would be better to clarify the advantages of the proposed model compared to these methods.
> > **[Question 2]** What's the advantage of the proposed model compared with diffusion model-based methods?
> >
>
> Thanks for making this important question. Similar to other temporal methods, CSDI treats spatio-temporal data as multivariate time-series and ignores spatial structure. PriSTI introduces diffusion models to spatio-temporal imputation. They use two separate attention modules to incrementally aggregate temporal and spatial dependencies. However, this design decouples the spatio-temporal context and then ignores intrinsic interactions between spatial and temporal dimensions. In addition, PriSTI applies linear interpolation to the time series of each node to initially construct coarse conditional information. However, in spatially missing data, this strategy cannot provide effective information.
>
> Notably, we have conducted comparative experiments with CSDI [1] and PriSTI [2], please refer to the global rebuttal.
>
> > **[Question 1]** In Figures 4 and 5, why is the MAE result of the SAITS ∞ in the PV-US and CER-E datasets?
>
> Thanks for your question. The temporal imputation methods, such as SAITS and BRITS, treat spatio-temporal series as multivariate time series. These models struggle to effectively learn from large-scale spatio-temporal data, which is the high-dimensional time series. Under the early stopping settings, both SAITS and BRITS often terminate training process early, resulting in extremely high MAE values. To more clearly compare other methods, we have truncated the results of BRITS and SAITS in Fig. 2 and Fig. 3. We have replaced "∞" with accurate MAE results in the revised version. Please refer to Fig. 2 and Fig. 3 in the rebuttal-submitted PDF.
>
> **References**
>
> [1] Tashiro Y, Song J, Song Y, et al. Csdi: Conditional score-based diffusion models for probabilistic time series imputation[J]. Advances in Neural Information Processing Systems, 2021, 34: 24804-24816.
>
> [2] Liu M, Huang H, Feng H, et al. Pristi: A conditional diffusion framework for spatiotemporal imputation[C]//2023 IEEE 39th International Conference on Data Engineering (ICDE). IEEE, 2023: 1927-1939.

---

### Official Review · Reviewer_Ctvj · 2024-07-13

**Soundness:** 3
**Presentation:** 2
**Contribution:** 3
**Rating:** 5
**Confidence:** 4

**Summary:**

This paper leverages the Probably Approximately Correct (PAC) theory to study the message-passing mechanism of spatial-temporal imputation. Inspired by the results of PAC, this paper introduces a One-step Propagation and Confidence-based Refinement (OPCR) model for spatial-temporal imputation. OPCR is comprised of a spatial and a temporal sparse attentions, and a confidence-based refinement module. Experimental results on several benchmark datasets show that OPCR could outperform several baselines.

**Strengths:**

1. Using Probably Approximately Correct (PAC) theory to analyze spatial-temporal imputation is an interesting direction.
2. Spatial imputation is an interesting sub-topic of the spatial-temporal imputation.
3. Experimental results on several benchmark datasets show that the proposed OPCR could outperform baselines.

**Weaknesses:**

1. In general, the proposed method is a little bit incremental. A simpler version of sparse spatial-temporal attention has been proposed by SPIN.
2. The presentation needs further improvements, especially the theoretical part. Many details and claims need clarification, please see questions.
3. The comparison baselines GRIN (2021) and SPIN (2022) are a little bit outdated. More recent baselines should be considered, such as [1,2,3].

[1] PriSTI: A Conditional Diffusion Framework for Spatiotemporal Imputation, ICDE'2023
[2] Networked Time Series Imputation via Position-aware Graph Enhanced Variational Autoencoders, KDD'2023
[3] Provably Convergent Schrödinger Bridge with Applications to Probabilistic Time Series Imputation, ICML'2023

**Questions:**

1. What does "poly" mean in line 98?
2. Line 101-102 & 104-105, what does $\phi$ mean?
3. Line 106-107, why should the model need to have the ability to recover all ST points?
4. In Assumption 3.2, what do $B_d$ and $B_x$ mean?

---

> ### Author Rebuttal · Authors · 2024-08-07
>
> Thanks for your careful reviews and insightful comments. We greatly appreciate your feedback. Please see the below responses to your comments (see global response to your questions about more baselines and confused notations).
>
>
> > **[Weakness 1]** In general, the proposed method is a little bit incremental. A simpler version of sparse spatial-temporal attention has been proposed by SPIN.
>
> Thanks for raising this fundamental issue. We would like to clarify the differences from SPIN and emphasize the contributions of the proposed method.
>
> **Differences.** The proposed method differs from SPIN in the following aspects:
>
> - SPIN propagates information from neighboring spatial-temporal points to the target ST point, while our sparse attention mechanism makes full use of global observations (i.e., fully connected graph) to recover every missing data, which may capture more information.
> - SPIN propagates in an iterative manner, while we directly propagate observations to missing data without any intermediary, which avoids the information loss and high computational cost caused by iterative propagation.
> - SPIN applies the sparse attention mechanism only in shallow layers. In deep layers, it assumes that most missing data has been filled and then employs dense attention mechanisms, which may lead to error accumulation, especially in large-scale datasets with high sparsity. Instead, our method only uses observations to recover all missing data in the first stage, ensuring consistent and accurate handling of sparse data.
>
> **Contributions.** Our contributions might not have been well received, and we restate and emphasize them. Specifically, our contributions lie in the theoretical analysis of imputation methods, as well as the breakthrough of theory-inspired methods.
>
> - **Our first contribution is in theoretically analyzing which factors impact the performance of the imputation task.** Most existing imputation methods usually rely on iterative message-passing in the temporal and spatial dimensions. However, their feasibility is based on their experience in engineering practice, and there is a lack of theoretical analysis. To this end, this paper aims to bridge the gap between theoretical analysis and empirical practice, and provide a theoretical analysis for general spatio-temporal iterative imputation models from the PAC-learnability perspective. The theoretical results reveal the impact of data sparsity and structural sparsity, as well as the need for more iterations for model performance. Notably, PAC-learnability-based analysis is not coupled to the specific model, and such general insights can inspire improvements to various models.
> - **Our second contribution is the imputation method OPCR that we have developed**, which has been shown to outperform the baselines regarding point missing and spatial missing tasks (Sec. 5.3 & 5.4). This is achieved by designing one-step imputation to avoid iterative error accumulation, and modeling spatio-temporal dependencies to refine the imputation results with confidence. Although the proposed method seems simple (i.e., reducing the number of layers and considering propagation over the fully connected graph), it is carefully designed based on comprehensive considerations of theoretical results, spatio-temporal dependence modeling, and computational complexity. We believe our method is quite useful.
>
> We reasonably believe that these contributions to the theoretical analysis of existing work and the algorithmic breakthroughs inspired by the theory are significant.
>
> > **[Weakness 3]** The comparison baselines GRIN (2021) and SPIN (2022) are a little bit outdated. More recent baselines should be considered, such as [1,2,3].
>
> Thanks for the valuable suggestions. Per your suggestions, we have added more baselines for comparison, including the tabular imputation method GRAPE [4], the time-series imputation method CSDI [3], the spatio-temporal imputation methods PoGeVon [2], and PriSTI [1]. Please see the global rebuttal.
>
> **References**
>
> [1] PriSTI: A Conditional Diffusion Framework for Spatiotemporal Imputation, ICDE'2023.
>
> [2] Networked Time Series Imputation via Position-aware Graph Enhanced Variational Autoencoders, KDD'2023.
>
> [3] Csdi: Conditional score-based diffusion models for probabilistic time series imputation. NeurIPS'2021.
>
> [4] Handling Missing Data with Graph Representation Learning. NeurIPS 2020.

---

> > ### Comment · Reviewer_Ctvj · 2024-08-14
> >
> > Thanks for the responses.
> >
> > My major concerns about (1) comparison with SPIN and (2) comparison with recent baselines are mostly addressed in the rebuttal. I'll update my scores later during the discussion period.

---

> > > ### Comment · Area_Chair_sNGY · 2024-08-14
> > > **Thanks**
> > >
> > > To Reviewer Ctvj,
> > >
> > > Thanks to your reply.
> > >
> > > best
> > >
> > > Area Chair

---

### Author Rebuttal · Authors · 2024-08-07

We sincerely thank all reviewers for taking the time to review our work. We appreciate that you find the problem is interesting and important (**Reviewer #Ctvj, #aMi7**), theory-inspired method is proposed (**Reviewer #g4JR, #aMi7**), the method is novel and efficient (**Reviewer #fc8s, #g4JR**), theoretical analysis is innovative (**Reviewer #Ctvj, #fc8s, #g4JR, #aMi7**), the paper is well organized (**Reviewer #g4JR**), with outstanding experimental results (**Reviewer #Ctvj, #fc8s, #aMi7**). Due to the limited space, the global response will answer the common questions, and the individual response will answer special questions.

**Q1: Concerns about more baselines.**
> **Reviewer #Ctvj**
> [Weakness 3] The comparison baselines GRIN (2021) and SPIN (2022) are a little bit outdated. More recent baselines should be considered, such as [1,2,3].
> **Reviewer #fc8s**
> [Weakness 2] Some diffusion model-based methods, such as CSDI [4] and PriSTI [1]
> **Reviewer #aMi7**
> [Weakness 5] Some of the related studies are highly related, and thus should be included. For example, [2, 5].

Following your suggestions, we have added more baselines for comparison, including the tabular imputation method GRAPE [4], the time-series imputation method CSDI [3], the spatio-temporal imputation methods PoGeVon [2], and PriSTI [1]. We present the imputation performances (in terms of MAE) of all methods in the METR-LA dataset with a missing ratio of 95% in the table below. It can be seen that the proposed OPCR is still significantly competitive on the two imputation tasks.

**Table 1. Imputation performance (in terms of MAE) of more baselines on METR-LA dataset.**
| Types | Methods | Spatial Missing | Point Missing |
| --- | --- | --- | --- |
| Tabular Data Imputation | GRAPE [4] | 6.78 | 6.73 |
| Time-series Imputation | CSDI [3] | 4.30 | 3.86 |
| Spatio-temporal Data Imputation | PoGeVon [2] |   10.14 | 9.47 |
| Spatio-temporal Data Imputation | PriSTI [1] | 5.01 | 3.90 |
| Spatio-temporal Data Imputation | **OPCR** | **4.22** | **3.15** |

**Q2: Concerns about confused notations.**
> **Reviewer #Ctvj**
> [Weakness 2] The presentation needs further improvements, especially the theoretical part. Many details and claims need clarification, please see questions.
> *[Question 1] What does "poly" mean in line 98?*
> *[Question 2] Line 101-102 & 104-105, what does 𝜙 mean?*
> *[Question 3] Line 106-107, why should the model need to have the ability to recover all ST points?*
> *[Question 4] In Assumption 3.2, what do 𝐵𝑑 and 𝐵𝑥 mean?*
> **Reviewer #aMi7**
> [Weakness 1.2] The notations for the theory provided in Section 3 do not offer detailed explanations.

We apologize for the confusing claims and notations. The detailed explanations of these claims and notations are as follows:
- **"poly":** The term “poly” in Definition 3.1 represents a polynomial function.
- **“$\phi$”:** The “$\phi$” denotes the activation function, which is assumed to be $C_{\phi}$-lipschitz continuous and bounded by $[0,1]$.
- **"Recover all ST points":** We apologize for the writing typos. The model needs to have the ability to recover all missing data, which is the goal of imputation tasks. We have clarified this point in the revised version.
- **"$B_d$, $B_x$":** For matrix $X$, we use $\lVert X\rVert_2$ to denote its spectral norm. For vector $x$, we use $\lVert x \rVert_2$ to denote its Euclidean norm. For any ST point $(v, t)$ in spatio-temporal dataset, we assume its collected feature vector $x_{v,t}$ satisfies $\lVert {x}_{v,t} \rVert_2 \leq B_x$. For the weight matrix $W_d$ in the decoder (line101-line102), we assume $\lVert W_d \rVert_2 \leq B_d$.
- **"$\mathcal{T}$":** Given a spatio-temporal series, we denote the set of nodes by $\mathcal{V}$ and the set of time steps by $\mathcal{T}$.
- **"$\mathcal{Z}_m$", "$\mathcal{Z}_o$":** We denote the set of all ST points as $\mathcal{Z}$. $\mathcal{Z}_o$ represents the set of all observed ST points and $\mathcal{Z}_m=\mathcal{Z} \backslash \mathcal{Z}_o$ represents the set of all missing ST points.
Thanks for your careful review. We have addressed each of your questions and have made corresponding modifications to the paper to improve clarity.


[1] PriSTI: A Conditional Diffusion Framework for Spatiotemporal Imputation, ICDE'2023.

[2] Networked Time Series Imputation via Position-aware Graph Enhanced Variational Autoencoders, KDD'2023.

[3] Csdi: Conditional score-based diffusion models for probabilistic time series imputation. NeurIPS'2021.

[4] Handling Missing Data with Graph Representation Learning. NeurIPS 2020.

---

### Decision · Program_Chairs · 2024-09-25

**Decision:**

Accept (poster)

**Comment:**

The paper presents  a novel framework for spatio-temporal imputation, leveraging Probably Approximately Correct (PAC) theory to offer a unique perspective on error accumulation in iterative imputation models. The theoretical analysis is innovative, particularly in its application to sparse attention-based imputation models. The authors propose a one-step propagation and confidence-based refinement framework, which effectively addresses error accumulation and can be applied to both spatial and temporal missing data scenarios. This high-level approach offers new insights for the design of future imputation algorithms. The paper also explores the challenges of spatially sparse data, emphasizing the importance of utilizing available data efficiently and lowering barriers to implementing spatio-temporal models in real-world applications. Experimental results on several benchmark datasets demonstrate that the proposed OPCR framework outperforms existing baselines across various spatio-temporal tasks. The theoretical analysis of existing iterative message-passing methods is highlighted as a significant contribution, potentially inspiring future research in sparse data learning. Additionally, the paper is well-organized, clearly identifying the limitations of existing methods through theoretical analysis and motivating the proposal of an improved method. The focus on tackling high sparsity in spatio-temporal data is particularly relevant, given its importance in many downstream applications. Overall, the paper combines strong theoretical foundations with practical experimental validation, making it a valuable contribution to the field of spatio-temporal imputation. The paper should be publised as NeurIPS2024.